# Interpolating numerically exact many-body wave functions for accelerated molecular dynamics

Yannic Rath [1,2] ✉ & George H. Booth [2] ✉

While there have been many developments in computational probes of both strongly-correlated molecular systems and machine-learning accelerated molecular dynamics, there remains a significant gap in capabilities in simulating accurate non-local electronic structure over timescales on which atoms move. We develop an approach to bridge these fields with a practical interpolation scheme for the correlated many-electron state through the space of atomic configurations, whilst avoiding the exponential complexity of these underlying electronic states. With a small number of accurate correlated wave functions as a training set, we demonstrate provable convergence to near-exact potential energy surfaces for subsequent dynamics with propagation of a valid many-body wave function and inference of its variational energy whilst retaining a mean-field computational scaling. This represents a profoundly different paradigm to the direct interpolation of potential energy surfaces in established machine-learning approaches. We combine this with modern electronic structure approaches to systematically resolve molecular dynamics trajectories and converge thermodynamic quantities with a high-throughput of several million interpolated wave functions with explicit validation of their accuracy from only a few numerically exact quantum chemical calculations. We also highlight the comparison to traditional machine-learned potentials or dynamics on mean-field surfaces.

The quantum fluctuations of interacting electrons represent the critical interaction between atoms which underpin all atomic bonding, dynamics, and reactivity. Computational approaches for systems with strongly interacting electrons have undergone a number of major developments in recent decades, as emerging methods enable a description of correlated electronic structure for ever larger and more realistic systems with unprecedented accuracy[1–3]. These modern approaches across both chemical and materials science include those based on tensor networks[4–8], stochastic methods[9–12], selected configuration interaction[13–15] and machine-learning-inspired wave function ansatze[16–25]. This has allowed for the near-exact solution to the quantum many-electron problem in these systems, providing high-accuracy insights for a few fixed atomic configurations, but have in general had little or no impact on our understanding of the physics and chemistry on the timescales of atomic and molecular motion.

The reasons for this are obvious; while a small number of single-point calculations with fixed nuclei are possible, the different timescales of atomic dynamics and electronic quantum fluctuations mean that on the order of at least thousands of sequential electronic structure calculations are required. This is essential to propagate the atoms in molecular systems to relevant timescales, entailing generally prohibitive computation expense for these high-accuracy methods. This is particularly challenging for these emerging methods which can lack a "black-box" use, requiring care to ensure reliable convergence at each

[1]National Physical Laboratory, Teddington, UK. [2]Department of Physics and Thomas Young Centre, King's College London, London, UK. ✉e-mail: yannic.rath@npl.co.uk; george.booth@kcl.ac.uk

point, while often also lacking analytic atomic forces to propagate the nuclear coordinates in time[26]. Important developments have been made in recent years in extending the application of established ground-state quantum chemical models to atomic dynamics[27–32], while "active space" methods are also increasingly widely used for stronger correlation or excited state molecular dynamics[33]. However, the additional cost of these approaches has meant that "ab-initio Born-Oppenheimer molecular dynamics" (AI-BOMD), where the atoms are classically propagated according to the potential energy surface of the electrons, is almost synonymous with a more empirical density functional description of the electronic structure which lacks systematic improvability and has many well documented deficiencies[34–36]. These include an over-stabilization of delocalized electronic states, as well as often inaccurate descriptions of dispersion forces, transition states, or bond-breaking among others[37]. These are critical parts of the phase space in real chemical dynamics, and the acute need for more reliable potential energy surfaces which build on the developments in accurate electronic structure is clear.

The most widespread and successful resolution to this need has come from a machine-learning approach to force fields[38–41]. These interpolate across chemical space between accurate single-point estimates of the electronic energy, based on local descriptors of the environment of each atom[42]. While this approach to straddling the electronic and atomic timescales has been arguably one of the most successful contributions of machine learning to quantum-level simulations to date, it is not without its own drawbacks[43]. In particular, the local nature of the descriptors can lead to difficulty describing long-range interactions[44], as well as "holes" where non-variational inferred energy estimates can lead to a collapse in the statistical sampling of phase space to these unphysical minima[45]. On a more fundamental level, since these approaches integrate out the electronic structure, there is in general no fundamental electronic variable at each sampled point (such as the wave function), meaning that the electronic properties which can be extracted are limited to the ones which correspond to the model definition. If the evolution of e.g., the dipole moment or charge distribution across a trajectory was desired this would not be accessible from a force field, and extensions to non-adiabatic effects are also far from straightforward in this framework, noting however significant recent research in these directions[46–49].

We take a different perspective and show that rather than interpolating observables such as the potential energy, we can instead interpolate the many-body electronic wave function itself through the phase space of molecular conformations. Importantly, despite the many-body wave function of each training point being in general exponentially complex, inference of properties from the model can be achieved in a scaling which is the same as (hybrid) density functional theory, rendering this a practical scheme. This decouples the unfavorable scaling of high-accuracy single-point electronic structure calculations from the evaluation of the interpolated potential energy surface, and thus allows for the use of these electronic structure methods for molecular dynamics on realistic timescales. We show that the resulting potential energy surfaces and molecular dynamics are systematically improvable to near-exactness via interpolation between highly accurate training configurations. Since a valid correlated many-electron state is propagated through the sampled phase space, this paradigm enables all electronic properties of interest to be simultaneously accessible within the same model, without relying on local or low-rank descriptors. Furthermore, since the energy is computed as a rigorous quantum expectation value over this inferred state, it provides a fully variational potential energy estimate (precluding "holes") for all atomic configurations, allows for clear evidence of systematic improvability to exactness as the training set is enlarged, an inductive bias of the model away from poorly described regions of phase space, and simple access to analytic atomic forces of the model for efficient propagation of dynamics.

We combine this approach for interpolating wave functions with modern density matrix renormalization group (DMRG) methods, allowing convergence of the strongly correlated potential energy surfaces to near exactness within the employed basis[4,5]. We demonstrate that this can provide a fully correlated electronic description of reactive molecular dynamics beyond traditional parameterized or machine-learned force fields, and ensembles of thermalized trajectories for equilibrated quantities over time scales which would be inaccessible without this acceleration. We show this can result in qualitative differences in behavior for a number of proto-typical molecular dynamics simulations compared to both density functional and traditional machine-learned force field approaches[50]. Finally, we compute both thermalized expectation values from canonical emsembles and reactive high-energy dynamical trajectories on a near-exact potential surface for the Zundel cation, a key intermediate for the Grotthuss mechanism for hydrogen diffusion through aqueous solutions[29,51,52]. With explicit validation of the accuracy of the surface, we compare the dynamics to both density functional theory results and other quantum chemical methods for both structural and electronic quantities, highlighting marked differences which can result from the quality of the surface.

## Results and discussion
### Interpolating wave functions
We first consider how to interpolate a single many-electron wave function between two different atomic configurations. We assume that we have an exact (FCI) correlated many-electron state defined within an atom-centered basis set of $L$ functions, for a specific set of atomic coordinates $\mathbf{R}$[53]. This wave function is a linear superposition over exponentially many electron configurations (Slater determinants) spanning the Hilbert space, as

$$|\Psi(\mathbf{R})\rangle = \sum_{n_1, n_2, \ldots, n_L} C_{n_1, n_2, \ldots, n_L} |n_1, n_2, \ldots, n_L\rangle, \qquad (1)$$

where $C_{n_1, n_2, \ldots, n_L}$ is the rank-$L$ tensor of probability amplitudes over the electronic configurations, and $n_i$ indexes the four local Fock states of the $i$th orbital; either unoccupied, spin-up, spin-down, or doubly occupied with electrons for each orbital. In general, both the probability amplitudes, and the single-particle orbitals defining each electronic configuration $|n_1, n_2, \ldots, n_L\rangle$ will change with atomic configuration $\mathbf{R}$. However, we aim to represent an approximation to the correlated electronic state at a different atomic configuration (and therefore electronic Hilbert space) with the same tensor of probability amplitudes over these electronic states. We exploit the fact that the properties of the exact state are invariant to orthogonal rotations of the single-particle orbitals, but that the probability amplitudes themselves will vary with this choice. Therefore, to enable transferrability between chemical environments, we seek a choice of orbital representation in which the probability amplitudes of the exact many-electron state change least between atomic configurations of interest.

A plausible choice is a basis of local atomic-like functions, appealing to the fact that a large portion of the electronic fluctuations among atomic-local orbitals will remain qualitatively similar as atoms are moved by small amounts. Similarly, regions of similar chemical bonding will also have common features in their probability amplitudes defining e.g., covalent fluctuations between neighboring atoms[54]. However, for reasons which will become clear, we also require that the orbitals represent an orthonormal set for all atomic configurations. To ensure this, while (in a least-squares sense) optimally preserving this atomic-like character of the orbitals, we symmetrically (Löwdin) orthonormalize the atomic-orbital basis (see Methods), defining orthonormal "SAO" orbital sets for each atomic configuration[55,56]. We can then choose to interpolate the state (and all resulting properties) between atomic configurations without re-optimizing the many-electron state by simply transferring the

probability amplitudes, while ensuring the consistent SAO basis definition.

This simple approach is limited, since the many-body amplitudes will in general change as the atoms move. However, we can generalize the state while retaining a valid wave function by linearly combining probability amplitude tensors in this transferable SAO representation from a larger "training" set optimized at other atomic configurations. We then variationally optimize the relative contributions of each of the $N$ training states for any test atomic configuration. This is achieved in closed form as the diagonalization of a generalized eigenvalue problem in the basis of the training states. By projecting the Hamiltonian at the desired test geometry $\mathbf{R}$ into this many-body basis, we get

$$\mathcal{H}(\mathbf{R})\mathbf{X}(\mathbf{R}) = \mathbf{E}(\mathbf{R})\,\mathcal{S}\mathbf{X}(\mathbf{R}), \qquad (2)$$

with the eigenvectors, $\mathbf{X}(\mathbf{R})$, giving the amplitudes of the training states defining the interpolated wave functions at the test geometry, with inferred energy spectrum $\mathbf{E}(\mathbf{R})$. The electronic Hamiltonian of the test geometry, $\mathcal{H}(\mathbf{R})$, is found by projecting the Hamiltonian operator into the many-body basis defined by the fixed probability amplitudes of the training states. This can be found in compact form as

$$
\begin{aligned}
\mathcal{H}_{ab} &= \sum_{ijkl}\sum_{\mathbf{n}\mathbf{n}'} C^{(a)*}_{\mathbf{n}} C^{(b)}_{\mathbf{n}'} \langle \mathbf{n}|\hat{c}^{\dagger}_{i}\hat{c}^{\dagger}_{j}\hat{c}_{l}\hat{c}_{k}|\mathbf{n}'\rangle\, K_{ijkl}(\mathbf{R}) \\
&= \sum_{ijkl} \Gamma^{ijkl}_{ab}\, K_{ijkl}(\mathbf{R}),
\end{aligned}
\qquad (3)
$$

where $\mathbf{n}$ denotes the many-electron configurations in the SAO basis of the test geometry, with $|\mathbf{n}\rangle \equiv |n_1, n_2, \ldots, n_L\rangle$, and with $C^{(a)}_{\mathbf{n}}$ and $C^{(b)}_{\mathbf{n}'}$ the fixed SAO probability amplitudes of the training wave functions at atomic geometries $a$ and $b$ respectively. This definition therefore implicitly transfers the probability amplitudes between the Hilbert spaces of the training and test geometries. The $K_{ijkl}(\mathbf{R})$ tensor is the two-electron reduced Hamiltonian defined in the SAO basis of the test geometry (given explicitly in the "Methods" section) with second-quantized Fermionic operators acting in this basis as $\hat{c}^{(\dagger)}$ shown.

Since the Hamiltonian is a sum of only two-electron interactions, the contraction over the exponential many-electron configurations, $\mathbf{n}$, is performed for all pairs of training states to give the transition two-body density matrices $\Gamma^{ijkl}$ defined in Eq. (3). Crucially, since the training probability amplitudes are defined not to change with test geometry, this contraction is only performed once on the training states, rather than for each test geometry. The construction of the subspace Hamiltonian at a test geometry therefore only requires the $\mathcal{O}[L^4]$ contraction of Eq. (3), with only the $K_{ijkl}(\mathbf{R})$ term changing with test configuration. The overlap between the training states, $\mathcal{S}$ of Eq. (2), can similarly be precomputed during the training phase, as

$$\mathcal{S}_{ab} = \sum_{\mathbf{n}} C^{(a)*}_{\mathbf{n}} C^{(b)}_{\mathbf{n}}, \qquad (4)$$

due to the orthonormality of the SAO basis at all test geometries. This ensures that the overlaps of the training states do not change with geometry, despite the physical training wave functions changing with atomic rearrangements as they transform between Hilbert spaces. In this way, the exponential complexity of the many-electron states are completely avoided in the inference of wave functions at new test points, by representing the training states in the polynomially-compact tensors $\Gamma^{ijkl}$ and their overlaps. The inference of the model requires a computational scaling of non-iterative $\mathcal{O}[N^2 L^4]$ after the density matrices of the training states have been precomputed in the training stage—the same formal scaling with system size as traditional (hybrid) density functional theory. This scaling for evaluation of the model at test geometries could also be further lowered with factorizations exploiting the low-rank nature of the $\Gamma^{ijkl}_{ab}$ tensors[57–60].

The lowest energy eigenvector from the diagonalization of this Hamiltonian (whose dimensionality scales only with the number of training points and is independent of system size) defines the specific variationally optimal linear combination of training probability amplitudes for the state, which can subsequently be used to predict any electronic property at this test geometry. Due to the variationality we have the desirable properties that the inferred state at a point which coincides with a training geometry must necessarily be exact, as well as the fact that each additional training point must necessarily lower the inferred energy towards the exact electronic solution across all possible test atomic configurations, assuming linear independence. In this way, the method more closely resembles a reduced order method than a machine learning model, where we define the Hamiltonian in a subspace defined by a fixed set of many-body vectors taken from training wave functions at different geometries, yet both are useful viewpoints. Due to the fixed training amplitudes across geometries, as well as the variational optimization of the model, computing analytic atomic forces from the inferred state is also straightforward (see Methods), ensuring a particular relevance of this acceleration in molecular dynamics applications.

This approach also builds on the perspective of "eigenvector continuation" which was recently introduced in both nuclear physics and condensed matter lattice models, where an eigenstate is analytically continued to different parts of the phase diagram[61–68]. Even more recently this was extended to simple ab initio quantum chemistry applications, with a related scheme to the one proposed[54]. However, a crucial difference was the use of a non-orthogonal atomic basis, which necessitated evaluation of the test point Hamiltonian directly from the many-body states. This retained the exponential complexity of the many-body state for inference at each test geometry, which is avoided here. The method of Mejuto-Zaera et al. was therefore presented instead as an approach for quantum computers, where many-body unitary operations can be applied in polynomial complexity. In contrast, the SAO basis for the interpolation formally breaks this requirement, ensuring the approach is amenable to classical computation in the predictions at test points with tractable mean-field computational cost.

A simple example of the scheme is shown in Fig. 1, for the symmetric stretch of a chain of six hydrogen atoms, with up to three atomic displacements considered in the training set. The compressed intermediate representation of the overlaps and transition density matrices between the training states in the SAO basis are shown, enabling variationally optimal predictions across the whole potential energy surface as linear combinations of the many-body training basis transformed between geometries. The predictions are found to converge to near exactness for this system with only three training points, with a guarantee of smoothness on adiabatic surfaces, and exactness at any training point geometry.

This approach is invariant to translation and rigid body rotations of the chemical system, provided a consistent ordering of the SAO representation is maintained, which is straightforward to achieve. However, this is not trivial for atomic permutations or point group symmetries which would change the SAO ordering, and hence probability amplitudes of the state definition. Furthermore, in contrast with building a force field based on local descriptors, the inference requires the same dimensionality Hilbert space for the electronic state, necessitating that the training and prediction points are taken from the same sized system. This is a significant difference to force field approaches with local representations which allow for scaling the system size after training, ensuring a different scope of applicability to the proposed approach[42]. Future work will look to relax this constraint.

While the proof-of-principle in Fig. 1 demonstrates excellent accuracy with few training points, it is also interpolation within a simple one-dimensional phase space of geometries. We now compare to a far larger phase space, composed of averaging the errors in both

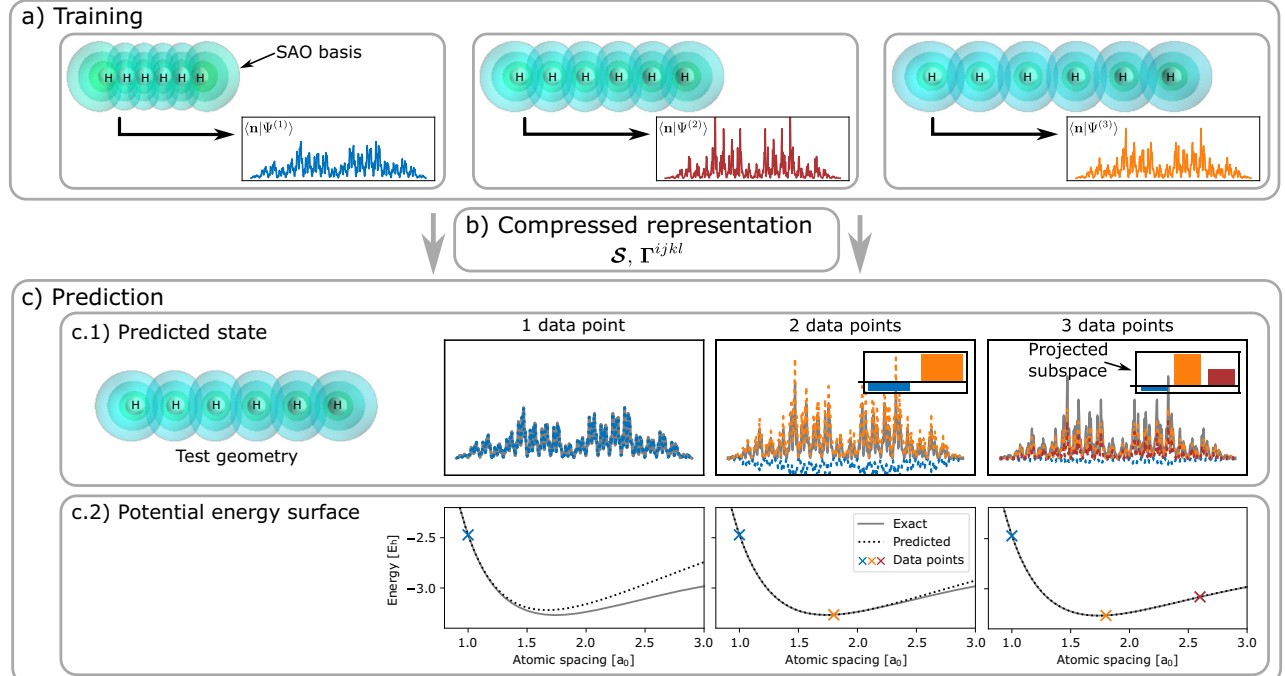

**Fig. 1 | Schematic overview of the proposed "eigenvector continuation" scheme for the high accuracy prediction across conformational space from few ab initio data points. a** Three many-electron training wave functions via exact diagonalization at different geometries in a Löwdin atomic orbital (SAO) basis of a linear 6-atom hydrogen chain. The values of $\langle \mathbf{n}|\Psi^{(a)}\rangle$ show the (exponentially many) wave function amplitudes for each training state. Geometry-agnostic one- and two-body transition density matrices ($\Gamma^{ijkl}$) and overlaps ($\mathcal{S}$) are constructed between all pairs of training states (**b**), which allows for fast variational prediction of the potential energy surfaces at arbitrary test geometries in the many-body basis of these training states (**c**). This allows efficient inference of wave functions at each test geometry, as shown in (**c.1**), with its associated ground state on the basis of the three training states. Plots in (**c.2**) show that enlarging the training space from one to three geometries systematically converges the full symmetric stretching mode of this system to the exact diagonalization result, with training data points where explicit electronic structure calculations are performed denoted by crosses. Source data are provided as a Source Data file.

the inferred energy and analytic forces on the atoms over randomly oriented three-dimensional displacements of each atom from a ten-atom linear hydrogen chain. This provides an exponentially large phase space of distorted chain configurations to test, where the radius of the displacements of each atom can be used to control the magnitude of the geometric distortions from the parent linear chain from which the training data is obtained. Only five training points from the symmetric stretch of the equidistant linear chain are used. We consider the increase in error as the magnitude of the displacements is increased in Fig. 2, as the test configurations move further from these training samples. We also compare these errors to a Gaussian approximated potential (GAP); a widely used machine-learning approach based on Gaussian process regression in a space of local descriptors from the superposition of atomic potentials[50,69]. This models a force field directly from the same training energies, but results in a materially larger error for the energy and forces over all displacements. We note that five points would generally be a very small training set for GAP, and that improved techniques to directly train on the forces of the training data themselves or improved model definitions were not used[70,71].

Nevertheless, a demonstration that inferring the wave function amplitudes themselves can outperform traditional machine-learning inference of the properties directly is noteworthy. Furthermore, we compare to Hartree–Fock theory (HF), which neglects all correlated electron effects and has the same computational scaling as the inference of the proposed "eigenvector continuation" scheme. This is also significantly worse at small distortions of the chain, though outperforms the largest distortions which are far from the training geometries and deep in the extrapolation regime.

## Bridging timescales

While it is easy to envisage many applications of an interpolation scheme for accurate correlated electronic structure, an obvious target is Born-Oppenheimer molecular dynamics (MD)[36]. In particular, the variationality of the scheme allows for systematic and quantifiable improvability to the exact solution of the electronic Schrödinger equation in the inferred potential energy surface at each geometry, while retaining a mean-field scaling with respect to the timescales which can be accessed. To access larger systems and basis sizes we also turn to modern electronic structure approximations for the evaluation of training states. In particular, we use DMRG to obtain training states with controllable accuracy to exactness[4,5]. These DMRG calculations can either be performed directly in the SAO basis or the state rotated into this basis after optimization, in advance of computation of the required overlaps and transition density matrices between the training states. As an alternative to DMRG, we are also able to approximate the training states by restricting the space of correlations to a low-energy complete active subspace (CAS) selected from the low-energy orbitals of a mean-field calculation[72].

We consider these approaches for constructing training states and the subsequent MD of a water molecule in increasing basis sets in Fig. 3. In particular, we consider convergence of the predicted vibrational frequency of the $a_1$ symmetric stretching mode as the number of training points increases. For the smallest basis, we find the full vibrational dynamics converge with just three training points, where we can compare directly to exact FCI calculations of the dynamics. As we increase the basis, FCI is intractable and we restrict the training to a CAS of low-energy orbitals, where the number of training points required grows modestly to seven and thirteen training points in a

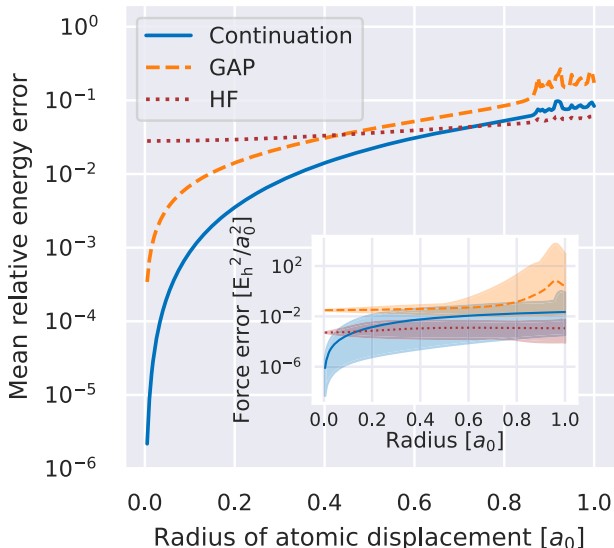

**Fig. 2 | Mean relative energy error of the prediction for distorted ten-atom hydrogen chains against the absolute displacement of each atom from the equilibrium position.** For each realization, a distorted chain was created by moving each atom from their position in the equilibrium geometry by a fixed displacement with a random direction. The comparison includes predictions from Hartree-Fock ("HF", red, dotted), the Gaussian approximation potential framework ("GAP", orange, dashed), as well as the variational continuation scheme from 5 training states of the symmetrically stretched chain ("Continuation", blue, solid). Each data point corresponds to the mean over 1000 randomly generated geometries. The inset shows the mean squared force error obtained with the three methods, where the shaded area denotes the range of the errors over the random realizations. The training set of equidistant one-dimensional geometries include the equilibrium length, with an interatomic distance of $\approx 1.79\,a_0$, as well as the 4 symmetric stretches of the atoms where the inter-atomic distance was increased and decreased by $0.5\,a_0$ and $1\,a_0$. Source data are provided as a Source Data file.

cc-pVDZ and cc-pVTZ basis respectively. We validate the specific trajectories found in Fig. 4, showing the difference between the inferred and reference energies at every time step. For the DMRG-based continuation in the 6-31G basis, we find that increasing the number of training states rapidly converges the full trajectory, with $N = 6$ training states achieving an accuracy well below $10^{-4}\,E_h$ across all points.

The variationality of the method guarantees that the predicted energies are always an upper bound to the exact ground state energy, at any geometry. When the continuation is based on approximate training wave functions, the inferred linear combination of training states may result in an improved energy compared to the reference, since it can mix contributions to the test state from other training geometries. This is even true when considering a geometry corresponding specifically to a training state. This is exemplified in the bottom panel of Fig. 4, detailing the energetic difference between the prediction and CASCI energies used for the training data along the trajectory in cc-pVDZ and cc-pVTZ basis sets. While the same active space sizes were used for the training states as for the computation of the reference energies, the inferred energies generally lie variationally below the CASCI reference energies. Although this improvement is small, mostly less than $1\,mE_h$, it is obtained for the majority of geometries over the converged trajectory, ultimately improving the accuracy beyond what is obtained with the reference method. As a further noteworthy difference between the reference CASCI and interpolated results based on CASCI training, for a small path of the trajectory between 24 and 25 fs in the cc-pVTZ basis of Fig. 4, a much more significant improvement of the continuation results over the reference method becomes apparent. This was found to be caused by a discontinuity in the CASCI ground state energies due to a change of orbitals included in the active space for these geometries. A

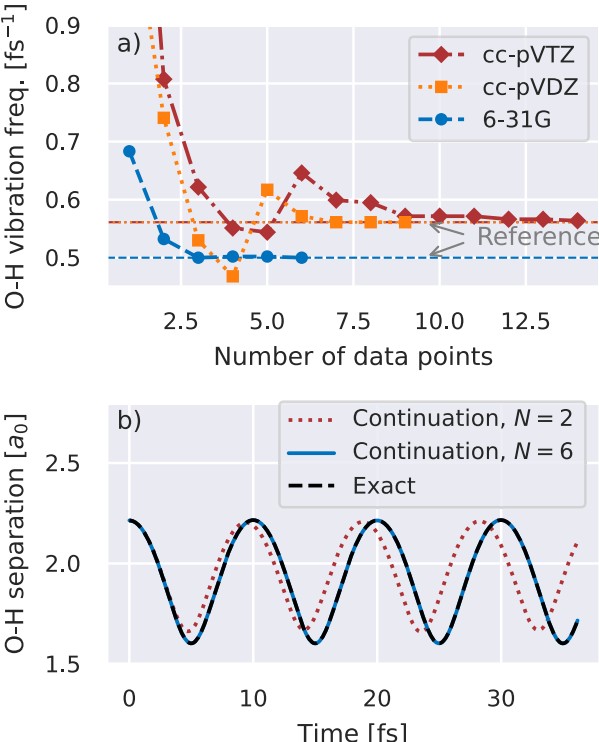

**Fig. 3 | Vibrational dynamics of a water molecule from molecular dynamics simulations with eigenvector continuation scheme. a** Predicted frequency of the $a_1$ symmetric stretch. Trajectories were started from a stretched initial configuration, and predicted with increasing numbers of training data geometries. We simulate the system in increasingly large 6-31G (blue, dashed), cc-pVDZ (orange, dotted), and cc-pVTZ (red, dash-dotted) basis sets where the larger two bases use training data restricted to a complete active space (CAS) of 4 electrons in 8 Hartree–Fock orbitals. Horizontal lines give reference values from trajectories on a FCI surface in the 6-31G basis, and CAS simulations in the cc-pVDZ and cc-pVTZ basis. **b** Oxygen-hydrogen distance over the trajectory in the 6-31G basis, as obtained from continuation with $N = 6$ (blue, solid) and $N = 2$ (red, dotted) training points, as well as the reference trajectory from full configuration interaction ("Exact", black, dashed). Source data are provided as a Source Data file.

more careful choice of active space is likely to have alleviated this problem in the reference trajectory, but we highlight it here since it is clear that this discontinuous change does not affect the interpolated surface. In contrast to the reference method on which it is trained, the continued results necessarily change smoothly with geometry, therefore mitigating a significant challenge in the use of active space methods in molecular dynamics.

It was found important to develop an active learning scheme for the selection of appropriate atomic configurations to include in the training data for rapid convergence. In ref. 54, the energy variance was motivated as an appropriate measure for the inclusion of data points, however this is impractical in the current lower-scaling scheme as it would require the evaluation of higher-body transition density matrices between training states. Instead, we consider the addition of training points which will maximize the improvement in the MD trajectories while respecting the invariances in the model predictions. This is performed by selecting the point on the trajectory where the Hamiltonian operator in the SAO basis, $K_{ijkl}(\mathbf{R})$, has changed most (in the least squares sense) compared to the Hamiltonians employed to generate the current training data set. Since the probability amplitudes are uniquely defined by this Hamiltonian, it is a suitable measure for the addition of new data points. Furthermore, due to the variationality of the method, it is guaranteed that the potential energy with the enlarged training data will be equal or lower to the previous

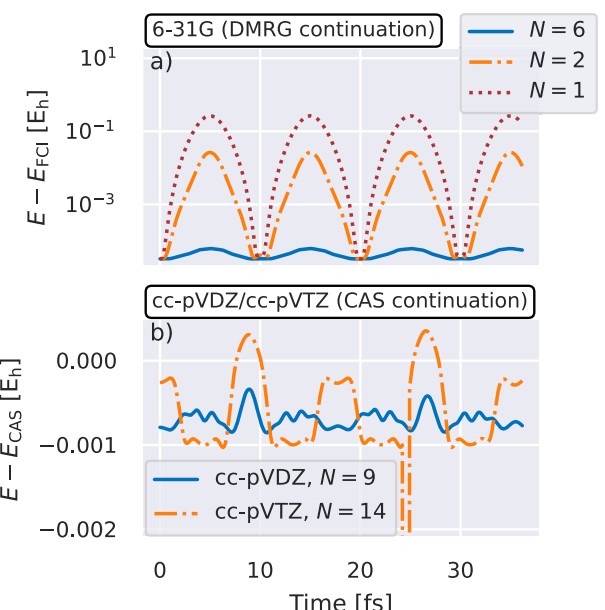

**Fig. 4 | Difference between the predicted energy and a reference method over all geometries from the dynamical trajectory of a water molecule. a** Error compared to exact diagonalization of density matrix renormalization group (DMRG) trained eigenvector continuation with increasing data set ($N$) in a 6-31G basis. **b** Difference between (4, 8) complete active space (CAS) trained eigenvector continuation and independently computed CAS energies at each geometry along the trajectory for $N = 9$ training points in a cc-pVDZ basis, and $N = 14$ training points in a cc-pVTZ basis. We stress that variationality with respect to this approximate training data is not expected, enabling the continued energies to be lower than the reference method, as shown. Source data are provided as a Source Data file.

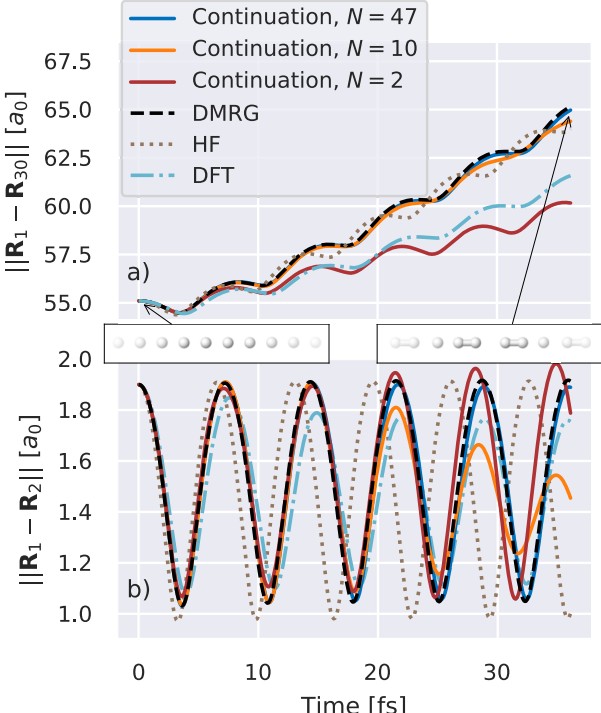

**Fig. 5 | Molecular dynamics of a 30 atom hydrogen chain in a STO-6G basis from an initially symmetrically stretched geometry.** The panels report the time-dependent Euclidean distance between two of the hydrogen atoms; **a** first and last atom in the chain, $\|\mathbf{R}_1 - \mathbf{R}_{30}\|$, **b** first and second atom, $\|\mathbf{R}_1 - \mathbf{R}_2\|$. This shows that the first two hydrogen atoms form a stable vibrating dimer while the overall chain lengthens. The trajectories were obtained from the eigenvector continuation ("Continuation") with $N = 47$ (blue), $N = 10$ (orange) and $N = 2$ (red) training points, together with the trajectories from density matrix renormalization group ("DMRG", black, dashed), Hartree-Fock ("HF", brown, dotted) and density functional theory with PBE exchange correlation ("DFT", light blue, dash-dotted) potential energy surfaces. Additional snapshots shown depict the initial and final hydrogen chain arrangements obtained from the converged eigenvector continuation ($N = 47$). Source data are provided as a Source Data file.

predictions, across the whole trajectory. This can therefore be used as a rigorous metric for the systematic convergence of the potential energy surface for the MD, with more details in the "Methods" section. We consider the potential energy surface over the whole MD simulation fully converged when the maximum reduction in energy for any point over the whole trajectory is less than 1 mE$_h$ for two consecutive increases in the data set size.

A semi-infinite symmetric one-dimensional chain of hydrogen atoms has emerged as a paradigmatic benchmark system of strongly correlated electronic structure in recent years, as a platform towards larger ab initio and extended systems. Almost all modern electronic structure methods have been applied to the system with varying success, and it has motivated further developments in both theory and understanding of its unexpectedly rich phase diagram[1,73]. While the symmetric stretch of this system has been considered extensively via single-point electronic structure, its full dynamics at this level have not. In Fig. 5, we release the atoms to dynamically move on a tightly-converged ground state surface (see Fig. S1 of the supplementary information for validation of the energy accuracy) of the DMRG-trained continuation scheme, starting from a ~ 10% symmetric stretching of thirty atoms equally from the symmetric equilibrium structure. We find that along with the vibrations of the bonds, the atoms rapidly dimerize and separate, with the overall length of the chain increasing approximately linearly with time. We are able to converge the dynamics of this dimerization and dissociation (albeit in a minimal basis) to the equivalent explicit DMRG AI-BOMD with only a small number of single-point training DMRG calculations. We note that in comparison, DFT-based AI-BOMD significantly underestimates the rate of dimerization of the chain, while Hartree–Fock theory conversely results in a bond for the hydrogen dimers which is too stiff, demonstrating the importance of an accurate treatment of the electronic correlations in the dynamics.

## Towards chemical accuracy for realistic (thermo)chemistry

We consider the feasibility of converging faithful thermodynamic quantities and reactive chemistry on a near-exact potential energy surface for the gas-phase dynamics of a Zundel cation, comprising a water molecule and hydronium ion—a system whose intricate potential energy surface poses a challenging test case for novel numerical techniques, yet is particularly important for the understanding of proton diffusion in aqueous solution[29,74–78]. We first consider a statistical ensemble of 500 different trajectories, starting from the same geometry (taken from ref. [79]), and sampling initial velocities from a Maxwell–Boltzmann distribution at 298.15 K. The BOMD was propagated under NVT conditions to thermalize according to a Berendsen integration scheme[80]. We consider $N = 60$, 80, and 100 single-point DMRG training configurations to observe the convergence of the thermodynamically equilibrated properties on a 6-31G basis. Each ensemble of trajectories at one of these training numbers involved $5 \times 10^6$ potential energy and force evaluations, which would be out of reach with a brute-force DMRG approach, but required a relatively modest 7500 CPU hours for the propagation of the full ensemble. Nevertheless, we can explicitly verify convergence to the accuracy of the underlying DMRG by a validation of the "test error" via additional DMRG calculations for sampled geometries along the trajectories. The achieved test error, shown in Fig. S2 of the supplementary information, demonstrates that the PES is well below chemical accuracy of the exact potential energy surface within the employed basis as the thermal

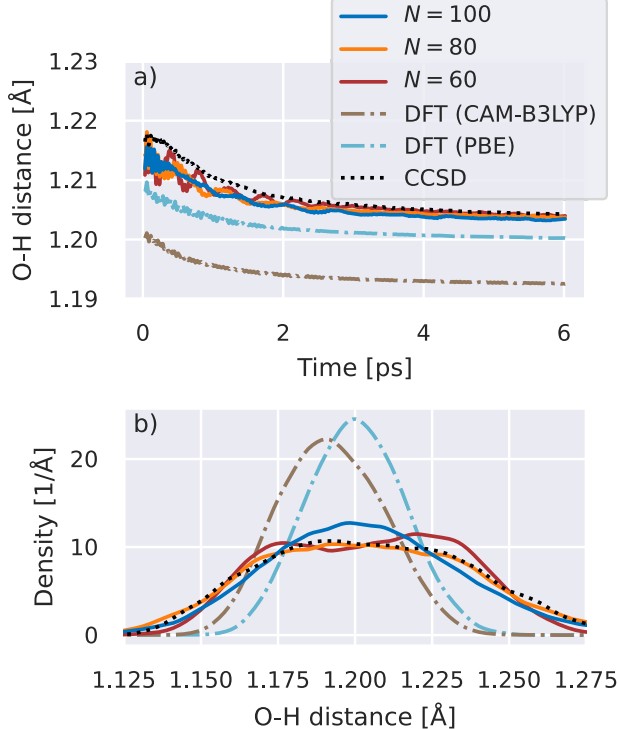

**Fig. 6 | Thermalized (298.15K) ensemble-averaged distance between the central hydrogen and the oxygen atom of the Zundel cation. a** Mean distance as a function of propagation time obtained by interpolating from $N = 60, 80, 100$ density matrix renormalization group training geometries, as well as results from density functional theory ("DFT" with CAM-B3LYP[103] and PBE[104] exchange-correlation functionals) and coupled-cluster with singles and doubles ("CCSD") trajectories. The mean corresponds to a running average of the distance between the atoms with a window of 100 timesteps ($\approx$60.4 fs), and averaging over 500 independent trajectories and both oxygen atoms. Each emsemble of trajectories required 5 million energy and force calculations. **b** Thermalized radial distribution function of the oxygen from the central hydrogen, using Gaussian smearing of individual data points in a kernel density analysis[105,106], with a bandwidth of $\sigma = 0.0025$ Å. Source data are provided as a Source Data file.

equilibrium is approached, reaching relative correlation energy errors below that of both CCSD and CCSD(T) – the "gold standard" of quantum chemistry[81].

Figure 6 shows this thermalization in the average distance between the central hydrogen atom and the two oxygen atoms in the explored Zundel configurations. We find this statistically equilibrated distance to be converging to a slightly shorter length than CCSD as the number of training configurations is increased. An accurate description of this multi-center bond is key for the Grotthuss mechanism of proton transfer. The differences in these quantities are in stark contrast to the much shorter distances predicted by DFT MD simulations with two widely used exchange–correlation functionals, which indicate a more localized central hydrogen. We can observe this in the radial distribution function of the equilibrated configurations (bottom panel) where the distribution is far flatter than the DFT methods, indicating an increased delocalization of the hydrogen between the water subunits. This is further corroborated by considering the magnitude of the dipole moment from the thermalized ensemble (see supplementary information, Fig. S3), which we find decreases as the level of theory is increased from DFT to CCSD to the DMRG-interpolated configurations, indicating a preference for more symmetric distributions where the central hydrogen is delocalized and less bound at any instant to an individual oxygen atom.

The verifiably high-accuracy interpolation coupled with the high-accuracy DMRG training allows for validation in the use of CCSD for

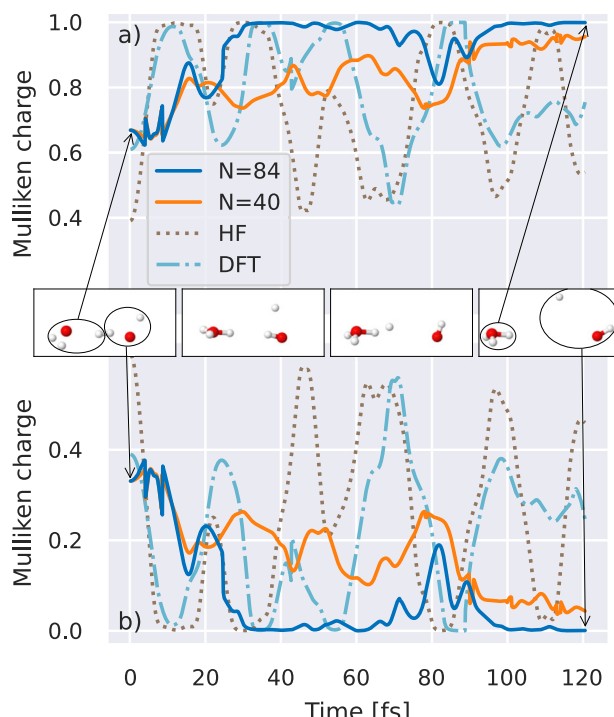

**Fig. 7 | Predicted Mulliken charge distribution for isolated high-energy molecular dynamics trajectories of a single Zundel cation.** Panels show the predicted Mulliken charges for the hydronium (**a**) and water (**b**) sub-units from a dynamical simulation of the reaction from eigenvector continuation with $N = 40$ (orange) and $N = 84$ (blue) training points from density matrix renormalization group. The system uses a 6-31G basis, with snapshots depicting the evolution of the molecular geometry at four evenly spaced times. Reference charges from simulations with density functional theory with a B3LYP exchange correlation ("DFT", light blue, dash-dotted) and Hartree–Fock ("HF", brown, dotted) are included for comparison. Source data are provided as a Source Data file.

this system, with thermalized expectation values qualitatively in agreement. This is due to the lack of strong correlation in the explored molecular configurations. However, a significant advantage of this framework is the ability to also reliably converge the PES over the full phase space, including strongly correlated atomic configurations further from equilibrium where CCSD is unreliable and will potentially fail, including bond-breaking and transition state geometries.

To consider this scenario, we also propagate a single high-energy trajectory within an NVE ensemble far from the Grotthuss mechanism dynamics, where the additional proton is inserted between the water monomers, interrupting the traditional hydrogen bond framework with a four-atom bridging bond as shown in the initial snapshot of Fig. 7. Increasing the number of DMRG training points to $N = 84$, we are able to observe convergence in the specific short-time MD trajectory over the 120 fs of the simulation (see Fig. S4 of the supplementary information for evidence of this convergence with training data over the trajectory). Due to the fact that an explicit representation of the electronic state is retained over the trajectory, we also extract non-energetic electronic properties of the system over time. We use this to consider the evolution of the Mulliken charge as the electron density is redistributed around the system in response to the atomic motion, beyond the physics considered in traditional polarizable force fields.

The positive charge is initially fairly evenly distributed amongst the water monomers, with the anticipation being that the water would rotate to adopt a lower-energy configuration. However, on the DMRG interpolated PES we find that before this is able to occur, a (neutral) hydrogen is ejected from the system leaving a bound

hydronium and hydroxide radical system. The charges on the sub-units of this reaction show the redistribution of charge as the hydrogen oscillates a number of times before its eventual ejection from the system. This behavior is not seen on the more approximate HF or DFT electronic surfaces where the additional hydrogen remains bound over timescales enabling the water to rotate its orientation, with the HF charge distribution substantially in error even in the initial state. Comparing to CCSD across the $N = 84$ trajectory, we find $\approx 40$ atomic configurations visited result in the CCSD energy diverging due to the presence of strong correlation effects, underlining the unreliability of the method for MD in these more unusual atomic conformations and transition states where strongly correlated electronic structure is found. This behavior is discussed further in the  supplementary information, but underlines the applicability of the continuation across the phase space of the MD and the potential to describe dissociative dynamics[68].

While these results demonstrate an effective acceleration scheme to converge the energy surface of this system to that of high-accuracy methods, more consideration of the effects of basis size, nuclear quantum and solvent effects may be needed before predictions as to the nature of this physical reaction can be given with confidence[29,30]. However, the fact that qualitative changes in dynamical behavior already result from the quality of the treatment of electronic correlation effects in the determination of the potential energy surface underlines the importance of a robust and systematically improvable approach to this electronic structure. The eigenvector continuation acceleration allows computation of this surface with high-level quantum chemical methods, and extends their scope to enable them to access timescales of atomic dynamics with provable convergence.

## Perspective

We develop a practical approach for eigenvector continuation of many-body electronic wave functions in ab initio settings. In contrast to the traditional paradigm of machine-learning force fields from training energies, this considers the interpolation of accurate wave functions across the space of structural changes, from which all electronic properties as well as atomic forces can be efficiently computed via a variational ansatz, avoiding the exponential complexity of the many-body states themselves. Using the scheme to converge the potential surface for molecular dynamics, we find examples of qualitatively different behavior to state-of-the-art techniques, demonstrating the importance of systematically converging the electronic structure across the timescales.

The acceleration scheme therefore holds huge potential to extend the scope of modern highly accurate electronic structure to molecular dynamics applications. However, the potential of reliable wave function interpolation also goes beyond this, towards a consideration of non-adiabatic and beyond-Born–Oppenheimer effects, efficient geometry optimization for ground, transition states or conical intersections, as well as a general procedure for vibrations and phonons, raising the possibility of the routine extraction of thermodynamic variables from accurate quantum chemistry. The move from single-point electronic internal energies to (thermo)dynamical quantities within correlated electronic structure theory is a long saught-after ambition[82]. The use of wave function interpolation with developments in solvers for the training data to extend system sizes could bring this closer to reality.

## Methods

### Interpolating across the ab initio potential energy surface

At the core of the methodology lies the prediction of the ground state electronic energy for given molecular arrangement of $N_{elec}$ electrons based on few exemplary solutions of the electronic structure problem at different molecular geometries. We define the ab initio electronic Hamiltonian for a $3 \times N_{nuc}$ atomic configuration, $\mathbf{R}$, in a discrete basis of electronic orbitals $\{\chi(\mathbf{r}; \mathbf{R})\}$ as[83]

$$\hat{H}(\mathbf{R}) = \sum_{ij} h_{ij}^{(1)}(\mathbf{R}) \, \hat{c}_i^\dagger \hat{c}_j + \frac{1}{2} \sum_{ijkl} h_{ijkl}^{(2)}(\mathbf{R}) \, \hat{c}_i^\dagger \hat{c}_j^\dagger \hat{c}_l \hat{c}_k + E_{nuc}(\mathbf{R}) \quad (5)$$

$$= \sum_{ijkl} K_{ijkl}(\mathbf{R}) \, \hat{c}_i^\dagger \hat{c}_j^\dagger \hat{c}_l \hat{c}_k + E_{nuc}(\mathbf{R}), \quad (6)$$

with Fermionic creation and annihilation operators, $\hat{c}^\dagger$ and $\hat{c}$ acting on the orbitals, and $E_{nuc}(\mathbf{R})$ the classical nuclear-nuclear repulsion energy. The one-electron terms, $h_{ij}^{(1)}(\mathbf{R})$, are matrix elements of the electron-nuclear and electronic kinetic operators, while the electron-electron repulsion integrals are

$$h_{ijkl}^{(2)}(\mathbf{R}) = \int \int d\mathbf{r}_1 d\mathbf{r}_2 \, \chi_i^*(\mathbf{r}_1; \mathbf{R}) \chi_j^*(\mathbf{r}_2; \mathbf{R}) \frac{1}{|\mathbf{r}_1 - \mathbf{r}_2|} \chi_k(\mathbf{r}_1; \mathbf{R}) \chi_l(\mathbf{r}_2; \mathbf{R}),$$
$$(7)$$

$$= \langle ij | kl \rangle (\mathbf{R}). \quad (8)$$

A convenient reduced two-body Hamiltonian which subsumes the one-body into the two-body term can be written as[84]

$$K_{ijkl}(\mathbf{R}) = \frac{1}{2} \langle ij | kl \rangle (\mathbf{R}) + \frac{1}{2(N_{elec} - 1)} \left( \delta_{jl} h_{ik}^{(1)}(\mathbf{R}) + \delta_{ik} h_{jl}^{(1)}(\mathbf{R}) \right). \quad (9)$$

The eigenvector continuation proceeds via the definition of a symmetrically (Löwdin) orthonormalized atomic orbital basis (SAO)[55,56]. This allows the training wave functions to be transferred between the Hilbert spaces of different geometries by fixing their many-body probability amplitudes in this representation. These SAOs are defined with an orbital transformation of an underlying non-orthogonal atom-centered "AO" orbital basis set at each geometry, $\{\phi_\alpha(\mathbf{r}; \mathbf{R})\}$, as

$$\chi_i(\mathbf{r}; \mathbf{R}) = \sum_\alpha [\mathbf{S}(\mathbf{R})]_{\alpha i}^{-1/2} \, \phi_\alpha(\mathbf{r}; \mathbf{R}). \quad (10)$$

where $\mathbf{S}(\mathbf{R})$ is the atomic orbital overlap matrix

$$S_{\alpha\beta}(\mathbf{R}) = \int d\mathbf{r} \, \phi_\alpha^*(\mathbf{r}; \mathbf{R}) \phi_\beta(\mathbf{r}; \mathbf{R}). \quad (11)$$

The continuation then proceeds according to the scheme outlined in the main text, with the evaluation of the transition two-body density matrices (t-2RDMs) and overlaps between the training points in their SAO representations. Of particular importance for molecular dynamics is the evaluation of analytic forces at each test geometry, which is simplified due to the lack of response contributions from the many-body basis and the fully optimized variational nature of the interpolated states in the geometry-independent basis[85,86]. This therefore only required the derivatives of the electron integrals in the AO basis[87–89], as well as the derivative of the transformation from the atomic orbitals to the SAOs with respect to nuclear positions (a "Pulay force"[90]), which we evaluate via first order perturbation theory[91]. The specifics of this evaluation is given in the supplementary information.

### Approximate training data: DMRG

Rather than relying on exact (FCI) training data, we also consider modern numerically efficient approximations to the correlated electronic structure to allow for access to larger systems, which are nevertheless systematically improvable to the exact solution to the electronic Schrödinger equation for training. These require not only the evaluation of accurate many-body wave

functions at the training geometries, but also the evaluation of the t-2RDMs and overlaps between different training states.

Firstly, we consider the compression of the training wave functions in the form of *Matrix Product States* (MPS), optimized via the *density matrix renormalization group* (DMRG) algorithm[92]. For this, we used the spin-adapted implementation from the *block2* library[5,93], working directly in the Fock space defined by the SAOs. We optimize the training states with a schedule for exponentially increasing bond dimension (a factor of 1.8 per increase) and decreasing noise in the MPS, a standard practice for stable ab initio DMRG[4], terminating when the difference of the energy upon fully relaxing the state at an increased bond dimension is less than a specified threshold, $\epsilon$. For all presented results, we set $\epsilon = 10^{-3}$ E$_h$ and start the MPS with an initial bond dimension of 34, giving training data confidently below the accepted "chemical accuracy". For the reference data for the hydrogen chain evolution (Fig. 5), we set the tolerance to $\epsilon = 10^{-5}$ E$_h$, and starting MPS bond dimension to 61.

**Approximate training data: CAS.** In addition to the continuation from MPS training states optimized with DMRG, we also present the use of *complete active space* (CAS) solvers to access the results of Fig. 3. These give an approximate ground state of the full electronic structure problem according to

$$|\Psi_{CAS}\rangle = |0\rangle^{N_{vir}} \otimes |\Psi_{AS}\rangle \otimes |1\rangle^{N_{core}}, \tag{12}$$

where $|\Psi_{AS}\rangle$ represents the fully variationally optimized state over all many-electron configurations within a chosen active subspace of orbitals and electrons, while $|1\rangle^{N_{core}}$ represents fully occupied orbitals spanning the remaining space of states that are occupied in a mean-field (in this case Hartree–Fock) description of the system, and $|0\rangle^{N_{vir}}$ explicitly indicate that the higher-energy virtual states are unoccupied. In this way, the electronic fluctuations of a low-energy subspace are considered fully, with the choice of active space in this work selected purely based on the mean-field orbital energies about the chemical potential of the system.

While this state can be straightforwardly optimized within a "CASCI" scheme implemented in the *PySCF* package[88,89], we also require the evaluation of the overlap and the t-2RDMs between training states in their SAO basis, while the state is defined (and optimized) in a mean-field canonical basis. Therefore, it is necessary to rotate these many-body states into their respective SAO bases before the t-2RDMs and overlaps are computed. We show this for the t-2RDM as

$$\Gamma_{ab}^{ijkl} = \langle \Psi_{CAS}^{(a)} | \hat{U}_{\mathbf{R}^{(a)}}^{\dagger} \hat{c}_i^{\dagger} \hat{c}_j^{\dagger} \hat{c}_k \hat{c}_l \hat{U}_{\mathbf{R}^{(b)}} | \Psi_{CAS}^{(b)} \rangle, \tag{13}$$

where $|\Psi_{CAS}^{(a/b)}\rangle$ denotes the CAS states at the different training points and $\hat{U}_{\mathbf{R}^{(a/b)}}$ is the unitary transformation from the state in the basis of its canonical orbitals to the SAO basis for the corresponding training point. This is evaluated efficiently as a double summation over the *active space* many-electron configurations (including their core) of each training state

$$\Gamma_{ab}^{ijkl} = \sum_{\mathbf{n}, \mathbf{n}' \in AS} C_{\mathbf{n}}^{(a)*} C_{\mathbf{n}'}^{(b)} \langle \mathbf{n} | \hat{U}_{\mathbf{R}^{(a)}}^{\dagger} \hat{c}_i^{\dagger} \hat{c}_j^{\dagger} \hat{c}_k \hat{c}_l \hat{U}_{\mathbf{R}^{(b)}} | \mathbf{n}' \rangle, \tag{14}$$

where $C_{\mathbf{n}/\mathbf{n}'}^{(a/b)}$ are the CASCI probability amplitudes of the active spaces. This single-particle unitary transformation $\hat{U}_{\mathbf{R}}^{\dagger}$ can be formed as

$$\hat{U}_{\mathbf{R}, ix} = \sum_{\alpha, \beta} Z_{\alpha i} S_{\alpha \beta} \tilde{Z}_{\beta x}, \tag{15}$$

where $Z_{\alpha i}$ is the transformation matrix from AO to SAO and $\tilde{Z}_{\beta x}$ is the transformation matrix from AO to canonical Hatree–Fock orbitals,

while $S_{\alpha\beta}$ is the AO overlap matrix. All of these quantities are dependent on the specific training geometry, $\mathbf{R}$.

The inner products $\langle \mathbf{n} | \hat{U}_{\mathbf{R}^{(a)}}^{\dagger} \hat{c}_i^{\dagger} \hat{c}_j^{\dagger} \hat{c}_k \hat{c}_l \hat{U}_{\mathbf{R}^{(b)}} | \mathbf{n}' \rangle$ from Eq. (14) can be identified as a matrix element between two different non-orthogonal Slater determinants[94]. The efficient evaluation of such overlaps between different non-orthogonal Slater determinants is discussed in refs. 95,96. We utilize the *libgnme* package, together with its python interface *pygnme*, to evaluate the overlaps and t-2RDMs between CAS states for the continuation in the SAO basis. Due to the non-orthogonality of the different CAS spaces, the double contraction of Eq. (14) results in a cost scaling quadratically in the size of the active space, thus more expensive than the evaluation of expectation values of a single point CAS state, however this cost could be reduced in the future by rotating to an intermediate basis representing the co-domain of the occupied CAS orbitals in a pair of CASCI training states.

**Gaussian Approximation Potentials.** We include comparison results obtained from the prediction of potential energies via *Gaussian Approximation Potentials* (GAP)[69]—a well-established framework for the prediction of potential energy surfaces and force fields. The model is extracted by fitting a data set of training geometries, $\{\mathbf{R}^{(a)}\}_{a=1}^{N}$, with associated energies $\{E^{(a)}\}_{a=1}^{N}$ using a kernel model[97] incorporating symmetries of atomic environments via the *smooth overlap of atomic position* (SOAP) descriptors[42]. We apply the GAP framework following standard approaches from the literature[50,70,98], based on the implementation of the SOAP descriptors in the *dscribe* package[99]. Additional details of this prediction procedure can be found in the supplementary information.

**Eigenvector continuation for BOMD**
The single-trajectory Born–Oppenheimer molecular dynamics of Figs. 3, 5 and 7 were computed in vacuum based on a microcanonical (NVE) ensemble using the Velocity-Verlet integration implemented in *PySCF*[36,88,89,100], according to the analytic nuclear gradients derived for the eigenvector continuation in the supplementary information. The nuclei in these simulations were initialized at rest, and we chose a fixed timestep of $\delta t = 5$ a.u. $\approx 0.121$ fs for the integration.

To extract a thermalized ensemble for the dynamics of the Zundel cation of Fig. 6, we included a room temperate (298.15K) Berendsen thermostat[80] as implemented in *PySCF* to obtain a canonical (NVT) ensemble of trajectories. This scheme relies on an additional rescaling of the velocities after each integration step to achieve an exponential convergence to the target temperature with a timescale $\tau$. Initial velocities for each trajectory were drawn from a Maxwell-Boltzmann distribution, while the nuclei positions were initialized in the ground state geometry obtained from CCSD(T) in a large basis set from ref. 79. The dynamics proceeded with a total of 10, 000 integration steps with $\delta t = 25$ a.u $\approx 0.605$ fs, and a thermalization time constant of $\tau = 250$ a.u. $\approx 6.05$ fs. This required $5 \times 10^6$ force calculations to propagate the ensemble of 500 trajectories over the 6 ps timescale considered.

**Active learning for data selection.** For the molecular dynamics applications, we perform an active learning scheme in which we identify and select suitable molecular configurations for training the eigenvector continuation scheme on-the-fly. This scheme is based on iteratively running the MD with a given training set, and selecting an enlarged training dataset with a new molecular configuration from the sampled trajectory. A correlated electronic structure calculation is performed at the selected geometry which is then included in the training data for an improved inferred potential energy surface and resulting MD trajectory in the next step. Starting from just a single training state (the initial geometry) and iteratively adding new configurations to the dataset in this way, the number of costly electronic structure calculations can be minimized and the trajectory can be

systematically and rapidly converged, noting that adding training geometries from the simulated trajectories guarantees an improved prediction in each step.

To select the new training geometry, we develop a "distance" heuristic for all geometries along the trajectory, which can be used as a metric for the addition of the data, and quantifies the suitability of the current training data in describing the test state at each point. The point along the trajectory with the largest measure is added to the training data set. Since the (non-degenerate) ground states along the trajectory are uniquely defined by the ab initio Hamiltonian at each geometry, we use the differences between the Hamiltonian elements at the training points and all trajectory points in defining this measure. Defining these elements in their respective SAO basis of each geometry used for the inference also ensures that the invariances and symmetries of the eigenvector continuation are also respected in this measure. Specifically, we define this Hamiltonian distance between two geometries, $D(\mathbf{R}, \mathbf{R}')$, as

$$
\begin{aligned}
D(\mathbf{R}, \mathbf{R}') = {} & \sum_{ij} |h_{ij}^{(1)}(\mathbf{R}) - h_{ij}^{(1)}(\mathbf{R}')|^2 \\
& + \frac{1}{2} \sum_{ijkl} |h_{ijkl}^{(2)}(\mathbf{R}) - h_{ijkl}^{(2)}(\mathbf{R}')|^2.
\end{aligned}
\tag{16}
$$

In addition to respecting the symmetries of the model, this ensures that two geometries with similar Hamiltonians (and thus wave functions) are considered similar, even though an evaluation of the Euclidean distance between these two geometries might be large (e.g., for geometries from near a dissociated limit). To add a new configuration, we evaluate $D(\mathbf{R}(t), \mathbf{R}^{(a)})$ for all geometries $\mathbf{R}(t)$ from the trajectory and each training geometry $\mathbf{R}^{(a)}$ already contained in the training set. We then pick that configuration $\mathbf{R}(t_{add})$ from the trajectory for which the distance to the closest training configuration is maximal, i.e., where

$$
t_{\text{add}} = \arg\max_t \left( \min_a (D(\mathbf{R}(t), \mathbf{R}^{(a)})) \right)
\tag{17}
$$

To gauge the systematic convergence of the NVE MD single-shot trajectories, we can track the variational lowering (and hence improvement) of the potential energy surface as the dataset is enlarged. This is done by comparing the PES from the two data set sizes along the same trajectory corresponding to the larger of the two data sets. Exploiting the variationality of the method, it is guaranteed that the potential energy inference with the larger dataset will be lower or equal to the predictions with the smaller dataset, and we use the difference between the predicted energies as a convergence measure. In our applications, we terminate the simulation when the predicted energy with the enlarged data set stays within a tolerance of $\epsilon = 10^{-3}$ $E_h$ along the full MD trajectory for two iterations in a row. Examples of this convergence are shown in Fig. S4 the supplementary information.

To manage the increased data volume when generating the statistical canonical ensemble of trajectories for the NVT Zundel cation results of Fig. 6, we use a somewhat coarser scheme to select the training configurations. We start with just the initial configuration in the training set, and randomly sub-sample 100 trajectories from the ensemble of 500 trajectories generated by the prior CAM-B3LYP DFT dynamics. For all timesteps of these 100 trajectories, the Hamiltonian distance metric of Eq. (16) is computed and the 19 geometries with the largest value of this metric are identified for inclusion in an enlarged training data set. The continuation scheme is then run for 500 NVT trajectories with these 20 training points. We compute the Hamiltonian metric along the full path of a new random selection of 100 of these inferred trajectories in order to identify a further set of 20 geometries to perform explicit DMRG calculations to iteratively enlarge the training data set until the desired size is reached. This

training set is taken to be the same for all trajectories in an ensemble. It should be stressed that only the first batch of 19 geometries are taken from the DFT-derived trajectories, after which subsequent batches of training geometries are found self-consistently to ensure a systematically reducing bias due to the DFT paths.

## Data availability

The raw data in this manuscript are provided in a Source Data file. The main molecular dynamics trajectories generated in this study have been deposited in the data repository at https://doi.org/10.5281/zenodo.14532437[101]. This repository also includes animated videos of the simulated nuclear motions in the NVE ensemble. Source data are provided with this paper.

## Code availability

The code and inputs to fully reproduce the numerical experiments of this work can be found at https://github.com/BoothGroup/evcont[102].

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

## Acknowledgements

We thank Huanchen Zhai for support with the *block2* code, Hugh Burton for support with the *libgnme* code, Carlos Mejuto-Zaera for helpful insights about an independently developed related scheme, Oliver Backhouse for technical help and Kemal Atalar, Venkat Kapil, and Lachlan Lindoy for additional feedback on the manuscript. G.H.B. gratefully acknowledges support from the Air Force Office of Scientific Research under award number FA8655-22-1-7011 and the UK Materials and Molecular Modelling Hub for computational resources, which is partially funded by EPSRC (EP/T022213/1, EP/W032260/1 and EP/P020194/1). Y.R. also acknowledges the support of the Engineering and Physical Sciences Research Council (EP/Y005090/1).

## Author contributions

Y.R. and G.H.B. jointly developed the methodology and wrote the manuscript. YR implemented the approach and performed the numerical experiments. G.H.B. supervised the project.

## Competing interests

The authors declare no competing interests.
