## [Transparent Peer Review file · Nature Communications]

Interpolating numerically exact many-body wave functions for accelerated molecular dynamics

Corresponding Author: Professor George Booth

This manuscript has been previously reviewed at another journal that is not operating a transparent peer review scheme. The manuscript was considered suitable for publication without further review at Nature Communications.

Version 0:

Reviewer comments:

Reviewer #1

(Remarks to the Author)

The authors propose a scheme to interpolate the electronic wave function between a number of nuclear geometries used as training sets. They demonstrate their methods in variants for a number of examples where they extract different properties of interest from the interpolated wave function.

While the present manuscript is mostly of methodical nature I consider it an important contribution.

However, two points should be addressed especially for their realistic chemistry model, the Zundel cation.

1) Observables that are interpolated with machine learning methods are usually calculated within milliseconds on test geometries, once the training is complete. On the other hand they need much more training data so that the training is numerically expensive. How do the timings compare to the method the authors propose?

2) If I understood correctly, the authors aim at calculating electronic properties along classical nuclear trajectories. The configurational space here is much smaller as compared to the case when the nuclei are quantum. Especially in Zundel the quantum effects of the nuclei are extremely important. Could their method also be applied here?

(Remarks on code availability)

Reviewer #2

(Remarks to the Author)

In their paper, the authors present an interpolation scheme of many-body electronic wavefunctions that allow the prediction of the electronic energy for geometries at which the parametric dependence of the energy was not included in the interpolation scheme. The idea and the methodology are interesting and innovative. Moreover, the soundness of the approach is justified by the exemplary applications that the authors provide, where the results show on test systems a fast convergence of the predicted energy for a small number of points and accurate reproduction of the physicochemical properties of these systems based on these interpolations. Having provided the code and extensive data is highly remarkable.

However, the authors propose that the approach introduced would challenge the state-of-the-art machine learning potentials, while similar limitations can be found between the two strategies in terms of transferability to different chemical systems, the generalizability of the approach, and the impossibility of obtaining accurate calculations in training to extend the approach with systems that drastically scale in the number of particles. One of the main messages of the manuscript is the acceleration of molecular dynamics, but unfortunately, this is not demonstrated for chemically relevant systems, and the methodology proposed, strongly relies on the need to calculate energies at several points of the conformational space of the system with accurate wavefunction. This does not really show yet potential applicability in a framework where an acceleration of

molecular dynamics would be reached.

For this reason, a recommendation for publication in Nature Communication could be advised only if the author could show how the approach would lead to the solution of some of these issues, for which the current examples are not satisfactory. Indeed they only show how correct the method is, but the paper does not leave any practical outlook on how the eigenvector continuations could be used on real chemically relevant systems, for which so high-level calculations are not feasible.

In particular, the following points should be considered in a revised version of the manuscript:

- In the idea of moving to more complex and relevant chemical systems, made of more particles, it is not clear which limitations would be faced in the applications of the approach. Is it how infeasible would be to calculate several training points with accurate wavefunctions? Is it the number of training points to reach convergence that would drastically increase? Is the complexity of the potential energy surface that would not allow it?

- The scaling of the method for increasing training states was not discussed carefully. It seems that matrix elements take M^4 , whereas the number of states would form the matrix so T^2 , so a total scaling of T^2M^4 ? And does the interpolated wavefunction require diagonalization of a T^2 matrix, which typically would take T^3 operations?

- In Fig. 2, the appropriate comparison could include Carr-Parrinello MD since that is common to use for minor distortion of the geometry. At a minimum, mentioning CPMD in the main text would be less misleading to the reader.

- Would the author advise using CASSCF, RASSCF, or better CASPT2 to properly include both static and dynamic correlations? That could represent an important application for the method, especially in the framework of dynamics. CAS-based dynamics is highly employed and reliable, being able to accelerate it by reducing the number of electronic structure calls using the approach proposed would be an important achievement.

- It is known that in CAS-based dynamics the stability of the active space is an issue, and preserving an active state along the dynamics is not guaranteed. In less problematic cases the approach proposed does not show any theoretical issue. But how do the authors think their method would perform on highly unstable active spaces?

- In Fig. 6, the instability of CAS-based dynamics is very pronounced. While the DMRG curves look sensible, CAS-based eigenvector continuation fails due to the inclusion of very disparate CAS spaces along the trajectory. When no crossing of BO surfaces is present, CASSCF with sensible choice (such as full valence space) should fix it. Alternatively, using automatic generation of valence active space (AVAS) might circumvent the issues without CASSCF. If crossing across different BO surfaces, more complexity is involved to interpolate between near the conical intersection of the surfaces.

- The increasing challenges when increasing the number of particles become clearer in the dynamics of the Zundel cation. Although the reason why DMRG dynamics were not run is clearly explained by the authors, the lack of dynamics obtained at the accurate DMRG level, the one used for the training, does not allow a chemical evaluation of the profile obtained by molecular dynamics. The authors should try to run AIMD at the same level of theory that they use for the training, to benchmark their method and make it trustworthy for future applications.

- In the same example, the authors could compare HF, DFT, and eigenvector continuation dynamics on an observable chemical property, such as free energy of dissociation or analogs, rather than solely charge values along the dynamics. This should be ideally run on an ensemble of trajectories.

- In Figure 7, could the author comment on why the difference in energy between predicted and DMRG oscillates so intensely? Although the accuracy of the relevant parameters computed along the dynamics is satisfying, is it correct to have a nonconstant error in the overall electronic energy? Would be interesting to see what is the correlation leading to this phenomenon.

- In cases of dissociations, the delocalization error (self-interaction error) is very common for DFT. A range-separated hybrid DFT is advised for a fair comparison. Perhaps that is the reason for the stable half charge of the hydronium at the end of the DFT dynamics in Fig. 5.

Additionally, minor points that should be considered:

- Through the text, the author uses the term "Chemical space", although in a misleading way. Indeed, this term is often used to indicate a space that composes all the possible chemical compositions of molecules, while in the manuscript is used to indicate different geometrical conformations of the same system. As one of the limitations of the approach is indeed the incapability to transfer the training obtained for a system to another one, a rephrase of the term chemical space is advised.

- Figures are very difficult to read. The font is very small and too many plots and labels are shown in a barely understandable gray.

- The title of the paper includes the term "near-exact electronic surfaces" which led me to think this paper is about conical intersections. Another phrasing such as "near-exact approximation of the electronic surface" might be less misleading.

- Deriving the forces for these eigenvector continuation states is a very important development (shown in the supplementary information). This is especially useful since it is made public in an open-source code that seems to be understated, and we encourage the authors to underline that achievement. Furthermore, these derivatives with respect to the overlap matrix are typically called "Pulay forces" in the quantum chemistry literature, and we advise the authors to use that term instead of the Hellman-Feynman theorem which does not have any wavefunction derivatives with respect to nuclear displacement.

(Remarks on code availability)

Reviewer #3

(Remarks to the Author)

(Remarks on code availability)

Reviewer #4

(Remarks to the Author)

Rath and Booth propose and implement a new method that greatly reduces the number of correlated electronic structure calculations e.g. during ab initio molecular dynamics simulations. The approach builds on the introduction of symmetrically orthonormalized atomic orbitals (SAO) for every geometry and subsequently interpolating the state (i.e. two body transition density matrices Γ_{ab}^{ijkl}). The method is in principle interesting and could be useful in a wide range of scenarios. Nevertheless, I have a number of concerns detailed below that preclude publication of the paper at the current stage:

1. the introduction correctly states the problems of density functional based AIMD but fails to present a balanced view of current challenges and contemporary developments in the field. For example, AIMD of condensed phase systems utilizing correlated electronic structure methods have been developed (see e.g. Hutter et al DOI: https://doi.org/10.1007/978-3-319-42913-7_58-1; Spura et al DOI: <https://doi.org/10.1039/C4CP05192K>). The introduction should reflect such recent advances.

2. The manuscript contains a number of inaccuracies, both, pertaining to language and notation, for example:

- Page 8: "realistic chemistry on the exact electronic surface" – the term exact electronic surface typically refers to converged data with respect to electronic structure method and basis set, but the presented data are for a limited basis set and probably far from convergence. The problems of simulations are somehow acknowledged on page 9 (l. 497-503), but not reflected in the section heading.

- Page 9: "interpolating across ab initio potential energy surfaces" implies multiple potential energy surfaces, like excited states which are not covered in the manuscript.

- Page 9, eq. 8: the R-dependence of 2-ERI. Similarly, eq.9 drops the R-dependence of K and 2-ERI, and $E_{\text{nuc}}(R)$ appears in eq. 5,6 but not in eq. 9 where it is explained. Such inaccuracies are particularly unfortunate, as the method is argued to benefit from the R-independent Γ_{ab}^{ijkl} used to interpolate between geometries (see below 3.)

3. transition two-body density matrices Γ_{ab}^{ijkl} are assumed to be independent of test nuclear configuration R which is at the core of the method for a common expansion of many body state. While the argument is made on page 3 "We can then choose to interpolate the state (and all resulting properties) between atomic configurations without re-optimizing the many-electron state by simply transferring the probability amplitudes, while ensuring the consistent SAO basis definition", I am not fully convinced by the argument. As the SAO basis is R-dependent, in principle expansion coefficients (eq. 1) and Γ_{ab}^{ijkl} all become R-dependent, unless orthogonality of SEO between nuclear configurations is assured. In the current approach orthogonality between SAO is only assured for a fixed R leading in principle to a R- (and thus t-)dependent moving SAO basis. The justification of the method should be worked out more thoroughly.

4. The relevance of the presented dynamics for H5O2+ is not fully convincing as no high level reference dynamics is presented. Deviations are recorded with respect to HF and B3LYP that both are prone to problems. Uncertainties of the dynamics are acknowledged to some extent on page 9 (l. 497-503) but no effort is undertaken to sufficiently corroborate the observations of the observed dynamics. For example (1) much more accurate and reliable functionals are available than B3LYP; alternatively (2) the PES of H5O2+ is known on a very high level of theory (see e.g. Huang et al DOI: <https://doi.org/10.1063/1.1834500>) and Born Oppenheimer dynamics on this surface should reproduce the current observations; (3) AIMD on the DMRG level of theory should be able to corroborate the method: 84 expansion points are already used, with the (very small) timestep 0.121 fs this amounts to ~10 fs propagation time on which deviations to HF and B3LYP are clearly recognizable. The argument that MD on DMRG level is not possible is thus not fully valid. To corroborate the observations, a more reliable reference has to be presented.

5. The chosen set of problems are more representative for gas phase quantum dynamics. As mentioned in the introduction, the black box nature of AIMD is particularly interesting for condensed, i.e., disordered liquid phase environments. What are the prospects of the proposed method upon such increased complexity?

(Remarks on code availability)

Version 1:

Reviewer comments:

Reviewer #1

(Remarks to the Author)

In my view the authors have addressed the concerns of all reviewers and the paper should be published after two minor issues are addressed:

Page 9, lines 569 "... where the additional proton is inserted between the ions...": please clarify "ions".

Fig. 6: In the inset I had trouble identifying the hydronium. The inset looks more like it is displaying a water dimer. Maybe the geometries could be rotated.

(Remarks on code availability)

The repository provides all information to run and install the code. The code is able to reproduce the data for the hydrogen chain. I was not able to install some of the dependencies that would allow me to reproduce the Zundel calculations. This, however, is probably due to my personal setup.

Reviewer #2

(Remarks to the Author)

The authors carefully considered all the points I raised in the initial revision, and satisfactorily answered, modified, and implemented changes in the manuscript. Especially the new experiment run by the authors completes the study and solves the initial flaws of the previous version of the manuscript. I now recommend publication in Nature Communications.

However, some figures are still showing very small font, it would be great if this could be enlarged before final publication.

Finally, in one of the answers, the authors assumed the gender of the reviewer to be male. Although it was only one occasion, it would be appropriate to carefully read the response before submitting it and ensure that such a legacy would not appear.

(Remarks on code availability)

Reviewer #3

(Remarks to the Author)

(Remarks on code availability)

Re: Response to reviewers comments on 'Interpolating many-body wave functions for accelerated molecular dynamics'

August 14, 2024

Dear Reviewers,

We would like to sincerely thank all the reviewers for the time they took to assess our work and their insightful feedback, and look to resubmit the manuscript in light of their constructive comments.

Guided by these, we have substantially modified the manuscript, in particular including a new large-scale numerical study on the Zundel system which considers an ensemble of trajectories to extract reliable thermalized quantities from the dynamics of our high-accuracy interpolated wave functions. Achieving this for provably accurate surfaces based around a novel acceleration scheme to enable high-throughput calculations of numerically exact DMRG approaches, we believe is a significant milestone and now demonstrates a broader application domain of relevant problems. We also provide further new results to explicitly validate the accuracy of this study compared to exact (DMRG) results within the chosen basis.

This new study considers a canonical (NVT) ensemble of 500 trajectories for the Zundel system in its equilibrium configuration, additionally extending the timescales of the dynamics by well over an order of magnitude. We demonstrate the approach to thermalization, and final statistical thermal quantities of relevance including the equilibrated oxygen-hydrogen distances and electronic properties in a canonical ensemble. This goes far beyond the previous study, substantially extending the number of training wave functions, and demonstrating the ability to compute over 5 million test points required for the ensemble of trajectories. The accuracy of the inferred potential energy surface generating this thermal distribution is explicitly validated by comparison to further benchmark DMRG calculations, demonstrating that the accuracy of the sampled surface significantly surpasses that of CCSD(T) and is able to be systematically controlled by the training data set size. We also compare beyond the PBE-DFT exchange-correlation functional, including a thermalized ensemble of CAM-B3LYP trajectories which constitutes a widely-used functional for this system as a counterpoint to PBE. Nevertheless, our distributions notably differ significantly from both of these DFT results.

Among other changes, we have also fully clarified the scaling to make it clear the applicability of the approach, as well as including additional results in the extended data detailing how the

interpolation ameliorates the problems with stability of active space choice which plague their traditional direct application to complex molecular dynamics.

Below, we include the reviews in full and address (in blue) each point raised, with reference to the relevant changes we have made to improve the manuscript. We provide a marked up version of this revised manuscript with changes highlighted in blue. We once again sincerely thank the reviewers for their time and consideration of this work, which we believe has resulted in a substantially improved manuscript, and look forward to your further communication.

Reviewer 1

The authors propose a scheme to interpolate the electronic wave function between a number of nuclear geometries used as training sets. They demonstrate their methods in variants for a number of examples where they extract different properties of interest from the interpolated wave function.

While the present manuscript is mostly of methodical nature I consider it an important contribution.

We thank the reviewer for their support of the work.

However, two points should be addressed especially for their realistic chemistry model, the Zundel cation.

1) Observables that are interpolated with machine learning methods are usually calculated within milliseconds on test geometries, once the training is complete. On the other hand they need much more training data so that the training is numerically expensive. How do the timings compare to the method the authors propose?

The reviewer raises an important point. In its current form, we would certainly suggest that the proposed approach is complementary to traditional interpolated force field ‘ML-FF’ methods, rather than something that is in direct competition with this approach. In particular, as mentioned, the speed to evaluate ML-FF models once training data is obtained is far faster than the continuation scheme proposed. The evaluation of our model scales as $\mathcal{O}[N^4]$, so more akin to hybrid DFT (albeit non-iterative), however this obviously significantly (perhaps exponentially) reduces the scaling of the correlated wave function scaling used in its training. However, the fundamentally different design of the approach compared to ML-FF techniques ensures that there are a number of features which go well beyond the scope of ML-FF approaches to provide a different perspective on acceleration of correlated electronic structure. These include its variationality, inference of true wave functions (and hence all electronic observables), multistate

inference and conical intersections (as we have recently demonstrated for application to non-adiabatic MD in <https://arxiv.org/abs/2403.12275>), fast convergence with training data in the 'low-data' limit, use of 'global' features for long-range physics and electrostatics, and ability to treat strong correlation physics in a consistent framework.

Regarding the practical cost of the scheme, our new study required 5 million force and energy calculations to propagate an ensemble of trajectories for the thermalized Zundel system (500 trajectories requiring 10,000 timesteps each), which extended the timescales of previous results by over an order of magnitude. With 100 DMRG training wave function, this required 7,500 CPU hours to infer the energies and forces to propagate all trajectories in this ensemble in our current (non-optimal) implementation (noting that this did not include the time for the DMRG evaluation of the training states, or the fact that the trajectories are propagated multiple times in the active data selection process – see Methods for more details). By way of comparison, performing DMRG (in the same SAO basis) to converge the correlation energy for training required at each point was ~ 12 CPU hours per geometry, which very roughly would put the cost of propagating the ensemble at 60 million CPU hours – a speed up of over three orders of magnitude. We hope this numerical evidence demonstrates that this is not just a method of academic curiosity, but can genuinely be a tool for realistic molecular dynamics – especially in scenarios which can exploit one of the features of the method that would render ML-FF more challenging as discussed above.

We have clarified the scaling of the approach in the paper, and included these relatively modest computational resources in the description of the Zundel dynamics.

2) If I understood correctly, the authors aim at calculating electronic properties along classical nuclear trajectories. The configurational space here is much smaller as compared to the case when the nuclei are quantum. Especially in Zundel the quantum effects of the nuclei are extremely important. Could their method also be applied here?

This is a particularly interesting research direction. Approaches like path-integral or ring-polymer MD require even more evaluations of the potential energy, and would be an ideal application domain for this acceleration scheme for accurate surfaces (along with non-adiabatic MD). Tunneling rates are particularly sensitive to barrier heights, and so high levels of accuracy for the potential energy surface are required, which is the particular strength of this approach, as well as correctly describing the (often strongly correlated) transition states. We have recently extended the approach to non-adiabatic MD with multiple electronic surfaces (see above), and believe that tunneling rates and quantum nuclei would be another ideal application area in the near future. We however think that this is currently outside of the scope of this initial paper focused on BOMD, but mention it in the manuscript as both an important physical mechanism to describe for accurate Zundel dynamics, as well as a direction of current research.

Reviewer 2

In their paper, the authors present an interpolation scheme of many-body electronic wavefunctions that allow the prediction of the electronic energy for geometries at which the parametric dependence of the energy was not included in the interpolation scheme. The idea and the methodology are interesting and innovative. Moreover, the soundness of the approach is justified by the exemplary applications that the authors provide, where the results show on test systems a fast convergence of the predicted energy for a small number of points and accurate reproduction of the physicochemical properties of these systems based on these interpolations. Having provided the code and extensive data is highly remarkable.

We thank the reviewer for these positive comments. We would however like to just ensure that there is clarity in the comment 'the parametric dependence of the energy was not included in the interpolation scheme'. There are indeed a number of aspects which introduce a parametric dependence of the inferred energy on the atomic geometry. Firstly, the Hamiltonian changes at each 'test' point. Furthermore, the coefficients of the model which describe the particular linear combination of probability amplitudes at each point are also specifically optimized for each inference point. The training *wave functions* also change with nuclear test geometry due to the associated SAO atom-centered basis changing. The parametric *independence* alluded to, which is crucial for the efficiency of the scheme, is that the *probability amplitudes* of the training states are independent of geometry (which are then linearly combined in a geometry-specific fashion).

However, the authors propose that the approach introduced would challenge the state-of-the-art machine learning potentials, while similar limitations can be found between the two strategies in terms of transferability to different chemical systems, the generalizability of the approach, and the impossibility of obtaining accurate calculations in training to extend the approach with systems that drastically scale in the number of particles. One of the main messages of the manuscript is the acceleration of molecular dynamics, but unfortunately, this is not demonstrated for chemically relevant systems, and the methodology proposed, strongly relies on the need to calculate energies at several points of the conformational space of the system with accurate wavefunction. This does not really show yet potential applicability in a framework where an acceleration of molecular dynamics would be reached.

We reiterate the response to reviewer 1, to clarify that we do not mean to claim that this approach would 'challenge' ML-FF approaches, but is rather a complementary approach which features a number of attributes which are very different to those in the ML-FF community and a different direction in which to consider interpolation of electronic structure. We believe that motivating the approach via a comparison to ML-FF was instructive as a way to highlight the differences as a counterpoint (e.g. full wave function interpolation, no atomic decomposition or selection of features/representations etc), but have now explicitly indicated in the text that as it stands this is not a 'replacement' for ML-FF approaches, which we did not intend to claim. Nevertheless,

we are, of course, exploring many further developments which aim to enlarge the scope of applicability of this approach towards this common aim of enabling high-throughput electronic structure.

Regarding the applicability to an acceleration of molecular dynamics, we hope that the new study, where longer-time thermalized ensembles of trajectories of the Zundel system are simulated with the approach, can persuade the reviewer of the applicability and relevance of the approach, even at this relatively early stage in its development compared to the maturity of ML-FF approaches. While still a relatively small system, these systems are representative of systems with wide relevance, and allows for the developments in correlated quantum chemical methods to be transferred to dynamics and high-throughput calculations in a consistent and controlled fashion. Furthermore, extensions of the method whereby the wave functions are appropriately 'fragmented' (in the manner of emerging quantum embedding approaches) raises the prospects of a more transferrable scheme and ability to scale up the number of particles in the system based on rigorously quantum chemical training. We allude to these possibilities of future developments in the text.

Regarding the comment as to the cost of the training, it is clear that future work will have to address this cost with available techniques to push towards larger systems. This will necessarily require more approximate correlated wave function methods which can nevertheless still treat strong correlation. We see the generality of the proposed framework as key for this, allowing for a wide scope in training wave functions and a hierarchy of trade-offs between system size and correlation treatment of the training data. This is an ongoing research direction (e.g. using CASSCF or other multideterminant training wave functions, selected CI, stochastic methods and even coupled-cluster for training data), but we certainly see no reason for pessimism that this would not be possible. However, we focus on a largely exact level of theory (apart from the CASCI training) in this work.

For this reason, a recommendation for publication in Nature Communication could be advised only if the author could show how the approach would lead to the solution of some of these issues, for which the current examples are not satisfactory. Indeed they only show how correct the method is, but the paper does not leave any practical outlook on how the eigenvector continuations could be used on real chemically relevant systems, for which so high-level calculations are not feasible.

As mentioned above, the substantially enlarged study obtaining thermalized ensembles of trajectories, we trust, demonstrates the applicability to realistic MD for the first time from a numerically exact DMRG level of theory, while the rigorous quantification of the associated errors demonstrate the fidelity of the electronic surfaces obtained. Furthermore, changes to the manuscript stress the complementary nature of the approach to ML-FF, but also highlights the differences in scope.

In particular, the following points should be considered in a revised version of the manuscript:

- In the idea of moving to more complex and relevant chemical systems, made of more particles, it is not clear which limitations would be faced in the applications of the approach. Is it how infeasible would be to calculate several training points with accurate wavefunctions? Is it the number of training points to reach convergence that would drastically increase? Is the complexity of the potential energy surface that would not allow it?

This is a good question, and one which is very hard to answer conclusively, and which will also depend on the specifics of the system. Of course, the first limitation is the ability to compute (an approximation to) the training wave functions for the model. In this work, we wanted to stress the ability to converge ‘exact’ quantum chemical models for the potential energy surface, which naturally is most limited in terms of system size. However, as mentioned above, it is clear that going to more approximate techniques will be necessary to scale up the system sizes. The framework can naturally accommodate this, and we are currently working on extensions for a variety of more approximate correlated electronic structure methods, with proof-of-principle results already for coupled-cluster, selected CI, and stochastic methods. The developments in this direction will synergistically depend on the developments in the active field of correlated electronic structure methodology, and allow for a wave function interpolation which complements the direct property interpolation of ML-FF communities.

Regarding the scaling of the number of training points, it is clear that the number of training points required will depend on the diversity of the phase space which is sampled in the dynamics (as it does with ML-FF), with e.g. the simple vibrations seen to converge very rapidly with just a few geometries. It is also clear that improved selection schemes could be devised to ensure that the number of training points can be kept as small as possible, as discussed in the extended data. The high-energy Zundel cation system certainly exhibits exploration of a large configurational phase space, and therefore its converged description provides some confidence that the approach is extendable as long as the training points can be found.

- The scaling of the method for increasing training states was not discussed carefully. It seems that matrix elements take M^4 , whereas the number of states would form the matrix so T^2 , so a total scaling of $T^2 M^4$? And does the interpolated wavefunction require diagonalization of a T^2 matrix, which typically would take T^3 operations?

All scalings stated here are correct, and we have modified the manuscript to further clarify these scalings. Two further points to mention however: T is generally small (a maximum of 100 in this work), as we are working in a ‘small-data’ regime (with each training point contributing information about a whole global wave function), therefore diagonalization of the $T \times T$ matrix is negligible in run-time. If we extend to the larger-data limit where this could become limiting, iterative diagonalizers or sparse data techniques could reduce this dependence, and so this is not significant. The bottleneck in the test phase of the model is therefore the $T^2 M^4$

scaling as you suggest. However, we have found recently that the matrices admit a low-rank form (as has been also explored in the literature), which can potentially reduce the scaling to M^3 – this is active work, which goes beyond the scope of this paper, but raises the prospect of further substantial increases in the accessible timescales in the near future. We also now include the actual computational time for the 5 million inference points for the ensemble of Zundel trajectories from our (far from optimal) development code, as discussed in the response to reviewer 1.

- In Fig. 2, the appropriate comparison could include Carr-Parrinello MD since that is common to use for minor distortion of the geometry. At a minimum, mentioning CPMD in the main text would be less misleading to the reader.

We have included a reference to CPMD as a more efficient approach for small distortions of DFT-based electronic potentials and AI-BOMD. We however expect these results to be equivalent to the explicit DFT-based AI-BOMD for the purposes of the paper.

- Would the author advise using CASSCF, RASSCF, or better CASPT2 to properly include both static and dynamic correlations? That could represent an important application for the method, especially in the framework of dynamics. CAS-based dynamics is highly employed and reliable, being able to accelerate it by reducing the number of electronic structure calls using the approach proposed would be an important achievement.

This is related to the discussion of alternative solvers – a research direction that we agree would represent an important and impactful contribution, and is currently being actively developed to extend the method beyond interpolation of ‘exact’ potential energy surfaces. Our initial results with CASCI training in the paper are interesting, in that the interpolation manages to go beyond the limitations of the active space employed in the training, effectively mixing character from nearby training geometries to variationally improve the interpolated state (see lower panel of Fig. 7 in the extended data, which uses CAS-CI). This is straightforwardly extendable to CASSCF. The reviewer correctly highlights the importance of dynamic correlation, however the suggestion of CASPT2 is a little more difficult to implement due to the perturbative correction and difficulty in defining the transition density matrices for this model. As an alternative however, we also have some preliminary results with coupled-cluster, which can be used to enlarge the tRDMs of a low-energy strong correlation method in a subspace to correct for dynamic correlation, and would increase the scope of the framework in the future. There is some more development work (and potentially additional approximations) required before this is ready, and therefore decided in this work to focus on the case where systematic improvability of the surface to exactness is possible before introducing and benchmarking additional approximations which we think would hinder the clarity of the paper. Nevertheless, we think these are exciting directions to further enlarge the scope of this framework.

- It is known that in CAS-based dynamics the stability of the active space is an issue, and

preserving an active state along the dynamics is not guaranteed. In less problematic cases the approach proposed does not show any theoretical issue. But how do the authors think their method would perform on highly unstable active spaces?

This is a very pertinent question, additionally related to our response to the next comment, where the reference CASCI trajectory exhibits exactly just such an instability. As mentioned, it is well known that CAS-based dynamics suffer from active spaces adiabatically transforming from relevant to irrelevant subspaces through the dynamics, which is a major drawback in their use. More dramatically, 'root-flipping' as states enter or leave the chosen active space along a trajectory can cause discontinuities which plague CAS-based AI-MD of strongly correlated systems, with often disastrous consequences for the resulting dynamics. Since this work considers every point on the surface as an optimal linear combination of the (fixed) probability amplitudes over all training points (and their associated CAS choices), then this approach does indeed offer the possibility of fully alleviating this problem, and rigorously removes discontinuities as the potential must be a smooth holomorphic function due to the linearity of the solution.

While the next response demonstrates this feature of the method which was already present in our data, we are also working on further work to show this more conclusively. We include some preliminary results from this further study in this letter (but not the manuscript) to further illustrate this. In the figure below (Fig. 1) we consider the torsion of an Ethylene molecule, with a (2,8) SA-CASSCF active space and aug-cc-pVDZ basis. This results in a spectrum with discontinuities (blue lines) as states enter or leave the active space resulting in abrupt change in all states in the spectrum. However, we find that training on just four geometries along this torsion, allows the eigenvector continuation to correctly and smoothly interpolate between these solutions both on the ground and excited states (red crosses and dotted lines), which should enable faithful molecular dynamics on these surfaces even when trained in cases of inadequate active space selection (work in progress). We also see the correct description of conical intersections which are generally not described with ML-FF approaches. While this is an exciting direction, we believe a full investigation of this phenomena and the scope of our interpolation to overcome it is not appropriate for this manuscript (which we have tried to keep accessible for a broad audience) but is being actively prepared for a follow-up manuscript. Nevertheless, we modify the manuscript to make this point, and we expand on these changes in the response to the next question below.

- In Fig. 6, the instability of CAS-based dynamics is very pronounced. While the DMRG curves look sensible, CAS-based eigenvector continuation fails due to the inclusion of very disparate CAS spaces along the trajectory. When no crossing of BO surfaces is present, CASSCF with sensible choice (such as full valence space) should fix it. Alternatively, using automatic generation of valence active space (AVAS) might circumvent the issues without CASSCF. If crossing across different BO surfaces, more complexity is involved to interpolate between near the conical intersection of the surfaces.

Figure 1: Alleviation of discontinuities found in CASSCF(2,8) (blue lines) via the eigenvector continuation (red dotted lines) on the electronic spectrum of Ethylene under torsion in an aug-cc-pVDZ basis. The continuation was trained on four geometries along the torsional coordinate (red crosses). [Plot for reviewers and not included in updated manuscript]

We thank the reviewer for carefully scrutinizing the results of our extended data and highlighting this – there is indeed an instability in the CAS spaces along the trajectory. As mentioned above and shown in Fig. 1, the continuation scheme precludes discontinuities in the potential energy surface due to the linearity of the model, even if trained on (necessarily approximate) wave function methods which exhibit discontinuities themselves. The plot in question (now Fig. 7 in the updated extended data) shows the error between the trajectory evaluated with CASCI and the interpolation based on $N = 9$ (VDZ) or $N = 14$ (VTZ) CASCI training points. The differences are on the order $\mathcal{O}[1 \text{ m}E_h]$, with the continuation being (in general) lower in energy due to the ability to mix character of the states between different geometries. The reviewer is right that there is a discontinuity in the error just before 25 fs in the plot – however crucially, this is due to the discontinuity in the CASCI energy *not* the interpolated surface on which the dynamics are run. We have highlighted this significant point to clarify this behaviour, and have added a comment in the main text to clarify that the CASCI-based continuation avoids the problems of

discontinuities, with an additional plot (the new Fig. 8) in the extended data to highlight the stability of the CASCI-based continuation trajectory compared to the 'parent' CASCI surface. We find that this discontinuity in the CASCI dynamics is removed in the interpolated data, and fully explains the jump in the interpolation error compared to the 'reference' CASCI results along the same trajectory.

- The increasing challenges when increasing the number of particles become clearer in the dynamics of the Zundel cation. Although the reason why DMRG dynamics were not run is clearly explained by the authors, the lack of dynamics obtained at the accurate DMRG level, the one used for the training, does not allow a chemical evaluation of the profile obtained by molecular dynamics. The authors should try to run AIMD at the same level of theory that they use for the training, to benchmark their method and make it trustworthy for future applications.

We agree that improved benchmarking of the approach was required, and have significantly improved the benchmarking of the accuracy of the dynamics of the Zundel system (see new Fig. 10). While we now have 5 million points for each ensemble of trajectories, which would be impossible for an explicit validation, we now choose an additional 1000 single-point DMRG calculations (converged to exactness) taken from equidistant times in the ensemble of trajectories (a validation data set which is ten times larger than the training data). A new plot in the extended data demonstrates the accuracy of the model over the trajectories from this validation set of 1000 geometries, also comparing to single-point CCSD and CCSD(T) energies at these points. We plot the relative correlation energy error over the trajectory, demonstrating that the thermalized ensemble of trajectories with $N = 100$ is converged to beyond the level of both 'chemical accuracy' and CCSD(T) for this system.

- In the same example, the authors could compare HF, DFT, and eigenvector continuation dynamics on an observable chemical property, such as free energy of dissociation or analogs, rather than solely charge values along the dynamics. This should be ideally run on an ensemble of trajectories.

In the updated manuscript we compute thermalized quantities from a statistical ensemble of 500 trajectories, both for structural parameters such as the oxygen-hydrogen distances and radial distribution function (Fig. 5 in the main body of the paper), crucial in the description of the subsequent Grothaus mechanism, as well as electronic properties (dipole moments) in the extended data, comparing between the DMRG-trained continuation dynamics, dynamics with DFT with two different exchange-correlation functionals, and coupled cluster with singles and doubles.

- In Figure 7, could the author comment on why the difference in energy between predicted and DMRG oscillates so intensely? Although the accuracy of the relevant parameters computed along the dynamics is satisfying, is it correct to have a nonconstant error in the overall electronic energy? Would be interesting to see what is the correlation leading to this phenomenon.

These ‘oscillations’ are easily explainable: given that the training data is essentially exact, the inferred energy should also be exact at that geometry. On a log-scale of the error, this shows up as sharp decreases in the error as you approach a training point, and increases away from it. The oscillations are essentially artifacts of the data selection process, indicating the proximity to the training geometries. We have clarified this in the updated manuscript to explain this phenomena.

- In cases of dissociations, the delocalization error (self-interaction error) is very common for DFT. A range-separated hybrid DFT is advised for a fair comparison. Perhaps that is the reason for the stable half charge of the hydronium at the end of the DFT dynamics in Fig. 5.

We have included a range-separated hybrid functional (CAM-B3LYP) in the extended study of the Zundel system for the thermalised distribution functions and dipole moments. However, we stress that the main message of the work is to present an alternative to the uncertainties that come with the choice of XC functional in DFT-based AIMD, by presenting an approach by which correlated wave function methods can be extended for use for MD. While inevitably closer agreement could be found with a wider exploration of different functionals, an important point of the manuscript is to circumvent this need. The additional results nevertheless indicate the scale of the variations which can be associated with different functional choices.

Additionally, minor points that should be considered:

- Through the text, the author uses the term “Chemical space”, although in a misleading way. Indeed, this term is often used to indicate a space that composes all the possible chemical compositions of molecules, while in the manuscript is used to indicate different geometrical conformations of the same system. As one of the limitations of the approach is indeed the incapability to transfer the training obtained for a system to another one, a rephrase of the term chemical space is advised.

This is certainly fair, and we have changed the use of the term ‘chemical space’ to alternatives, generally ‘conformational phase space’.

- Figures are very difficult to read. The font is very small and too many plots and labels are shown in a barely understandable gray.

We have updated figures to do our best to clarify the different lines, using additional (consistent) colour where possible to distinguish between methods and increasing font size of text on figures.

- The title of the paper includes the term “near-exact electronic surfaces” which led me to think this paper is about conical intersections. Another phrasing such as “near-exact approximation of the electronic surface” might be less misleading.

While the interpolation does approximate a number of low-lying states in the spectrum, with the extension to conical intersections and non-adiabatic dynamics described in the recent publication <https://arxiv.org/abs/2403.12275>, we agree that the title could be clarified for this work that focuses on ground state dynamics. We thought that the proposed change would make the title unnecessarily verbose, and as an alternative we have changed the title to ‘Interpolating many-body wave functions for accelerated molecular dynamics on the near-exact electronic surface’, to clarify that it is a single (ground state) surface guiding the dynamics.

- Deriving the forces for these eigenvector continuation states is a very important development (shown in the supplementary information). This is especially useful since it is made public in an open-source code that seems to be understated, and we encourage the authors to underline that achievement. Furthermore, these derivatives with respect to the overlap matrix are typically called “Pulay forces” in the quantum chemistry literature, and we advise the authors to use that term instead of the Hellman-Feynman theorem which does not have any wavefunction derivatives with respect to nuclear displacement.

We agree with the reviewer here, and thank him for his comments on the importance of open-source code – we certainly share this view, and have ensured that the inputs and scripts to fully generate the new results have also been included in the public repository for this work. We also agree that the derivatives with respect to the basis are called ‘Pulay forces’, and have indicated this in the updated manuscript. An important aspect however is that there is no (first-order) response due to the infinitesimal distortions of the wave function (either its amplitudes, or expansion coefficients) that needs to be considered in contrast to many other methods, or indeed the response of any underlying mean-field, which substantially simplifies the formulation.

Reviewer 3

Reviewer 4

Rath and Booth propose and implement a new method that greatly reduces the number of correlated electronic structure calculations e.g. during ab initio molecular dynamics simulations. The approach builds on the introduction of symmetrically orthonormalized atomic orbitals (SAO) for every geometry and subsequently interpolating the state (i.e. two body transition density matrices Γ_{ab}^{ijkl}). The method is in principle interesting and could be useful in a wide range of

scenarios. Nevertheless, I have a number of concerns detailed below that preclude publication of the paper at the current stage:

1. the introduction correctly states the problems of density functional based AIMD but fails to present a balanced view of current challenges and contemporary developments in the field. For example, AIMD of condensed phase systems utilizing correlated electronic structure methods have been developed (see e.g. Hutter et al DOI: https://doi.org/10.1007/978-3-319-42913-7_58-1; Spura et al DOI: <https://doi.org/10.1039/C4CP05192K>). The introduction should reflect such recent advances.

We agree that insufficient prominence was given to these important and relevant developments – while we noted and cited some of the work on MP2-driven AI-MD in passing, we have now explicitly acknowledged this work in the introduction and highlighted these contributions along with other relevant works.

2. The manuscript contains a number of inaccuracies, both, pertaining to language and notation, for example:

- Page 8: “realistic chemistry on the exact electronic surface” – the term exact electronic surface typically refers to converged data with respect to electronic structure method and basis set, but the presented data are for a limited basis set and probably far from convergence. The problems of simulations are somehow acknowledged on page 9 (l. 497-503), but not reflected in the section heading.

We agree that the exactness in this description refers to the basis set correlation energy, as described in the main text. We have changed the section title to ‘Towards chemical accuracy for realistic (thermo)chemistry’, and have also explicitly clarified the use of ‘exact’ in a number of other places in the manuscript as referring to the exact correlation within the employed basis set, to avoid confusion in this point. This also reflects the updated results of the section.

- Page 9: “interpolating across ab initio potential energy surfaces” implies multiple potential energy surfaces, like excited states which are not covered in the manuscript.

We have changed this to ‘Interpolating across the ab initio potential energy surface’ to indicate that we are currently only testing a single (ground-state) surface, noting that this was also highlighted by reviewer 2 in the choice of title for the manuscript (which has also changed). We also note in passing that the other roots of the Hamiltonian provide an interpolation for other states, which we explicitly use for a simple extension to non-adiabatic MD in recent work which we now cite in the paper (<https://arxiv.org/abs/2403.12275>).

- Page 9, eq. 8: the R-dependence of 2-ERI. Similarly, eq.9 drops the R-dependence of K and 2-ERI, and $E_{nuc}(R)$ appears in eq. 5,6 but not in eq. 9 where it is explained. Such inaccuracies

are particularly unfortunate, as the method is argued to benefit from the \mathbf{R} -independent Γ_{ab}^{ijkl} used to interpolate between geometries (see below 3.)

Yes, apologies. This explicit inclusion of the \mathbf{R} -dependence was dropped for aesthetic interests of keeping equations more compact, but was not done consistently. We have reintroduced the explicit \mathbf{R} -dependence of these quantities for clarity, and ensured that quantities are defined where they are first used.

3. transition two-body density matrices Γ_{ab}^{ijkl} are assumed to be independent of test nuclear configuration \mathbf{R} which is at the core of the method for a common expansion of many body state. While the argument is made on page 3 “We can then choose to interpolate the state (and all resulting properties) between atomic configurations without re-optimizing the many-electron state by simply transferring the probability amplitudes, while ensuring the consistent SAO basis definition”, I am not fully convinced by the argument. As the SAO basis is \mathbf{R} -dependent, in principle expansion coefficients (eq. 1) and Γ_{ab}^{ijkl} all become \mathbf{R} -dependent, unless orthogonality of SAO between nuclear configurations is assured. In the current approach orthogonality between SAO is only assured for a fixed \mathbf{R} leading in principle to a \mathbf{R} - (and thus t -)dependent moving SAO basis. The justification of the method should be worked out more thoroughly.

We do not think of the \mathbf{R} -independence of the t -RDMs an approximation, but rather a consequence of our ansatz definition. These (transition) density matrices are obtained from a projection into a well-defined variational subspace at each test geometry \mathbf{R} . This is done in such a way that ensures a) that it spans the space of the exact eigenstate when a test geometry coincides with a training geometry for which an exact solution was obtained (or with one related by a trivial symmetry), and b) the subspace can be systematically enlarged with increasing number of training states. As a crucial element of our framework, these training states are defined as fixed wave functions in the SAO basis, which can be evaluated for different test geometries. While the physical subspace of many-electron states spanned by the training states changes with \mathbf{R} , since (as you say), the SAO basis changes in space (but is nevertheless uniquely defined at each point), the wave function probability amplitudes of the training states in the SAO basis remain fixed. Consequently, the numerical values of the t -RDMs do not change and become independent of the nuclear geometry. With this perspective, the training states can be thought of as just defining an ansatz for a (many-body) subspace at each test geometry into which the Hamiltonian is efficiently projected and solved.

We require that the test point probability amplitudes are adequately described by an optimized linear combination of these fixed ‘training’ probability amplitudes. However, already with this definition, the t RDMs must remain \mathbf{R} -independent, as long as the basis is orthonormal. We select these training probability amplitudes from their local, orthonormal ‘SAO’ representations, and there is never any need to then consider the differential overlap between the real-space basis functions at different geometries, since the interpolation between many-electron states is done by interpolating the probability amplitudes across the SAO Fock space, and not across the

many-electron space of electronic real-space positions. We find that in this SAO representation, much of the local correlated fluctuations in the many-body state can be accurately obtained via their linear combination. In this perspective, the R -independence of the t-RDMs is then seen as not an approximation, but rather a consequence of our ansatz definition for the projector. Indeed, if we were to *not* neglect the differential overlap between the training states, then it would be easy to argue that the approach would not work, since the training wave functions would not 'move' with the test geometry under consideration, and e.g. electronic density of the training configurations would not be centered around the test geometry nuclei, rendering them an exceptionally poor choice of training data.

We can further justify qualitatively the use of an atomic-local representation in which to interpolate these training probability amplitudes, especially for capturing more strongly correlated physics, as the important quantum fluctuations in the probability amplitudes governing these are atomic-local in nature (e.g. d-shells of transition metals, or the suppression of atomic local double occupancy in a covalent bond or pair of stretched hydrogen atoms), which will remain similar under small geometry distortions. We appreciate that this justification can be quite unintuitive at first, and we have modified the manuscript a little to improve this justification further.

4. The relevance of the presented dynamics for H5O2+ is not fully convincing as no high level reference dynamics is presented. Deviations are recorded with respect to HF and B3LYP that both are prone to problems. Uncertainties of the dynamics are acknowledged to some extent on page 9 (l. 497-503) but no effort is undertaken to sufficiently corroborate the observations of the observed dynamics. For example (1) much more accurate and reliable functionals are available than B3LYP; alternatively (2) the PES of H5O2+ is known on a very high level of theory (see e.g. Huang et al DOI: <https://doi.org/10.1063/1.1834500>) and Born Oppenheimer dynamics on this surface should reproduce the current observations; (3) AIMD on the DMRG level of theory should be able to corroborate the method: 84 expansion points are already used, with the (very small) timestep 0.121 fs this amounts to 10 fs propagation time on which deviations to HF and B3LYP are clearly recognizable. The argument that MD on DMRG level is not possible is thus not fully valid. To corroborate the observations, a more reliable reference has to be presented.

As described in our responses to similar questions from other reviewers, we believe that we have comprehensively addressed these comments with our substantially extended study of the Zundel system, expanding the timescales and statistical ensembles of trajectories considered, comparing more extensively to other functionals (CAM-B3LYP) and levels of theory (CCSD and CCSD(T) – the same level as the Huang et al. paper referenced), and large-scale validation of the accuracy of the surface via many more DMRG calculations along the interpolated trajectory (see new Figs 5, 10, 11), demonstrating that we go beyond the accuracy of CCSD(T) for the thermally equilibrated dynamics. The propagation time for an ensemble of 500 NVT trajectories is extended to 6ps for the thermally equilibrated Zundel cation (5 million DMRG-

interpolated energies and forces per ensemble), while we also retain the (single) high-energy dissociative trajectory of this system to demonstrate the ability to converge strongly correlated dynamics in more unusual parts of the nuclear phase space where other methods can struggle even qualitatively. We hope this addresses these comments.

5. The chosen set of problems are more representative for gas phase quantum dynamics. As mentioned in the introduction, the black box nature of AIMD is particularly interesting for condensed, i.e., disordered liquid phase environments. What are the prospects of the proposed method upon such increased complexity?

This is of course an interesting direction in which we hope to extend the scope of this framework in the future. ‘Trivial’ answers would be that a) we could do so by explicitly considering the full system and be forced to work with cheaper more approximate training methods (e.g. selected linear combinations of determinants or even coupled-cluster) or b) include some electrostatic embedding (in both training and test hamiltonians) via a solvent model or point charge ‘MM’ framework, which would be straightforward to include. More interesting perhaps is whether an explicit electronic ‘environment’, including its hybridization with the system, could be treated in the interpolation framework. This is a direction we are actively pursuing, building on our experience with quantum embedding methodologies which naturally consider the appropriate way to fragment a wave function description of a system, while including the effect of its hybridization with other subsystems. We are hopeful of advances in this direction, but consider it too early a stage to include it in this manuscript. Nevertheless, this offers the prospect of being able to simultaneously straddle the length- and time-scales with high-accuracy quantum chemical methods by only requiring the training on subsystems.

Yours sincerely,

Yannic Rath and George Booth

January 27, 2025

Dear Reviewers,

We thank the reviewers sincerely for their help in improving the manuscript and getting it into the stage ready for acceptance. Please find below our response to the suggestions of final revisions by the reviewers.

Please let us know if there is anything else required from us.

Reviewer 1

Page 9, lines 569 "... where the additional proton is inserted between the ions...": please clarify "ions".

We have clarified this to 'water monomers', which more accurately describes the position of the additional proton.

Fig. 6: In the inset I had trouble identifying the hydronium. The inset looks more like it is displaying a water dimer. Maybe the geometries could be rotated.

We have rotated the geometries to make the additional hydrogen more easily identifiable.

Reviewer 2

However, some figures are still showing very small font, it would be great if this could be enlarged before final publication.

We have adjusted the styling of the figures, which included increasing the font size, to improve

the readability.

Finally, in one of the answers, the authors assumed the gender of the reviewer to be male. Although it was only one occasion, it would be appropriate to carefully read the response before submitting it and ensure that such a legacy would not appear.

We genuinely thank the reviewer for raising this, and are embarrassed that this slipped through, and deeply apologise for any offense. Please accept our apologies and we will certainly endeavor to ensure that this does not appear in the future.

Reviewer 3

No comments remaining to be addressed.

Yours sincerely,

Yannic Rath and George Booth